# Effects of nutritional interventions on nutritional and immunological status and adherence to antiretroviral treatment among adults living with HIV in low- and middle-income countries: Systematic review and meta-analysis

**Alain Nahaskida[1,2]\*, Jérome W. Somé[3], Befikadu Tariku Gutema[4], Nele S. Pauwels[5], Stefaan De Henauw[2], Souheila Abbeddou[2]**

1 Département de la Nutrition, Direction de la Nutrition et des Technologies Alimentaires, Ndjamena, Chad, 2 Department of Public Health and Primary Care, Faculty of Medicine and Health Sciences, Ghent University, Ghent, Belgium, 3 Centre National de la Recherche Scientifique et Technologique, Institut de Recherche en Sciences de la Santé, Unité Nutrition et Maladies Métaboliques, Ouagadougou, Burkina Faso, 4 School of Public Health, Arba Minch University, Arba Minch, Ethiopia, 5 Knowledge Center for Health Ghent, Ghent University and Ghent University Hospital, Ghent, Belgium

\* alain_nahaskida@yahoo.fr, alain.nahaskida@ugent.be

## Abstract

### Background

HIV/AIDS may cause malnutrition, both directly and indirectly, through common infections. This systematic review and meta-analysis aim to evaluate the effects of nutritional interventions on nutritional status, immunological status, adherence to antiretroviral treatment (ART), and food security among people living with HIV/AIDS (PLWHA) in low- and middle-income countries (LMICs).

### Method

Five databases—MEDLINE, Embase, Scopus, Web of Science, and CENTRAL—were searched for articles on August 17, 2021, with an updated search conducted on September 30, 2023 to identify new records. Studies were considered eligible if they included adults living with HIV/AIDS who recently initiated ART, if they were controlled trials that provided nutritional interventions, and if they assessed the relevant nutritional, immunological and adherence outcomes. The effects of nutritional interventions were analyzed using a random-effects model.

### Results

The systematic review comprised 22 articles from 12 LMICs, while the meta-analysis included 19 articles. The interventions provided lipid-based nutrient supplements, corn–soy blends, food baskets, conditional cash, prepared meals, micronutrient

**Data availability statement:** All included articles are published

**Funding:** This work was conducted under the PhD studies of AN, whose scholarship was funded by the International Atomic Energy Agency (IAEA, Vienna). The funders had no role in study design, data collection and analysis, decision to publish, or preparation of the manuscript.

**Competing interests:** No authors have competing interests

supplementation, and functional foods to PLWHA. Compared to controls, nutritional interventions for PLWHA significantly improved their body mass index (standardized mean difference, 95% confidence interval) (SMD 0.42; 95% CI: 0.03, 0.81; p = 0.03), fat mass (SMD 0.21; 95% CI: 0.07, 0.34; p = 0.002), fat-free mass (SMD 0.33; 95% CI: 0.19, 0.46; p < 0.0001), and CD4 (SMD 0.54; 95% CI: 0.01, 1.07; p = 0.05), but had no effect on their weight, viral load, or adherence to ART. The baseline nutritional and immunological characteristics of PLWHA, as well as the intervention characteristics, further modified these effects.

## Conclusion

Nutritional interventions improved some nutritional and immunological indicators but not ART adherence among PLWHA. Additionally, their effects were modified by some baseline characteristics and the type and duration of interventions which require consideration before its scaling up.

## Introduction

Malnutrition—ranging from undernutrition, which includes wasting, stunting, low body weight, and micronutrient deficiencies, to overnutrition and obesity—is affecting populations globally. Notably, the term malnutrition has been commonly used to define a state of undernutrition, which is also the synonym used in this paper. Malnutrition has continued to be a major public health problem in low- and middle-income countries (LMICs), disproportionally affecting children and women of reproductive age [1]. In adults, malnutrition results from poor dietary intake, increased energy expenditure, or disease-related reduced absorption, increased loss, or altered requirement for nutrients [2]. Human immunodeficiency virus and acquired immunodeficiency syndrome (HIV/AIDS) can cause malnutrition not only directly but also indirectly through common infections, such as tuberculosis, non-typhoid salmonellae, and *Streptococcus pneumoniae*, as well as opportunistic infections (e.g., those caused by *Leishmania* and *Tarolomyces marneffei*) [3].

HIV is a virus that attacks the human immune system, resulting in a weakened system that is unable to fight infections and diseases. Its routes of transmission are sexual contact, mother-to-child transmission, and exposure to blood. The HIV pandemic has claimed more than 40 million lives ever since it was first identified in 1981 [4,5]. Although HIV infections were on the rise in the 1990s and 2000s, they have remained a public health concern, especially among young adults [6]. The World Health Organization reported that the African region comprises two-thirds of people living with HIV [7]. Notably, the advent of the era of antiretroviral treatment (ART) saw a significant reduction in morbidity and mortality among people living with HIV/AIDS (PLWHA) [6]. ARTs are long-term medications used to manage HIV rather than to cure the infection. When adhered to properly, they can effectively suppress the virus and support the restoration of immune function. Patients who are able to control the virus are the only ones who regain immune protection and can maintain a healthy nutritional status [8]. HIV infection and malnutrition constitute a vicious cycle—one fueling the other. HIV and coinfections suppress appetite, increase the catabolism of muscles, and can result in wasting. Malnutrition, in turn, increases susceptibility to infections, boosts the replication of HIV, and can accelerate the progression of the disease to AIDS if the HIV-infected person is left untreated, ultimately leading to mortality. Moreover, malnourished PLWHA have high rates of treatment dropout and inferior response to treatment [1,9]. Malnutrition, defined

as the ratio of weight (in kgs) to the square of height (in meters)—also known as the body mass index (BMI)—being less than 18.5 kg/m2 in adults, is common among PLWHA (38% in Boston, US [10]; 43% in Salvador, Brazil [11]; and 77% in Iran [12]). In 1987, the Centers for Disease Control and Prevention defined HIV-associated wasting as involuntary weight loss >10% of the baseline body weight along with diarrhea, fever, or weakness for >30 days [3]. Currently, a more simplified definition of wasting is used, denoted as weight loss >10% or sustained weight loss >5% over a period of 6 months. The causes of HIV-associated wasting are categorized as follows: i) inadequate nutrient intake due to food insecurity, anorexia, diarrhea, malabsorption disorders, depression, and substance use; and 2) altered metabolism due to chronic infection, dysphagia, malignancy, hormonal imbalances, metabolic changes, and cytokine excess [13–15]. Although HIV-associated wasting is common in both high-income countries and LMICs, it reaches a more severe stage in food insecure settings.

Despite all the efforts to put an end to the HIV/AIDS pandemic, the prospect of achieving Sustainable Development Goal (SDG) 3.3, which aims to stop the HIV epidemic by 2030, is far from being achieved. HIV/AIDS disproportionally affects people residing in LMICs and those belonging to the poorest categories in high-income countries for whom the impacts of the disease are exacerbated by the challenges posed by food insecurity and social stigma. The prevalence of food insecurity among PLWHA is quite high. For instance, population studies conducted in the US reported that between 36%–50% of PLWHA were food insecure [10,16,17]. Similarly, a higher prevalence of food insecurity among PLWHA has been reported in studies from Kinshasa, the Democratic Republic of Congo [18], Kenya [19], and Ethiopia [20]. Furthermore, the prevalence of food insecurity among PLWHA was found to be as high as 70%–90% in some regions of Uganda [21], Senegal [22], and Namibia [23].

In the ART era, remarkable progress has been made in improving the quality and length of life of PLWHA, particularly in LMICs. In this context, when PLWHA are food insecure or experience weight loss, early nutritional intervention is vital for 1) maximizing the gain of lean body mass, 2) improving food security, 3) reducing the occurrence of opportunistic infections, and 4) supporting the immune system, thereby preventing decline and promoting rapid reconstitution [9,24]. As a result, achieving and maintaining optimal nutrition is considered a crucial strategy for both treating HIV/AIDS and ensuring food security for PLWHA. However, nutritional interventions conducted in LMICs have yet to be sufficiently examined to provide strong evidence to policymakers about the importance of combining nutrition with ART for better nutritional and health outcomes among PLWHA. This systematic review and meta-analysis aimed to analyze the effects of nutritional interventions on the nutritional and immunological status, adherence to ART, and additional outcomes, including hemoglobin, micronutrient status, and food security, of PLWHA living in LMICs.

## Methods and materials

The systematic review and meta-analyses conducted in this research have been reported in accordance with the Preferred Reporting Items for Systematic Reviews and Meta-Analyses (PRISMA) (S1 Checklist) [25]. After developing the protocol, it was registered with the International Prospective Register of Systematic Reviews (PROSPERO; registration number CRD 42021250824, published on July 4, 2021) before starting the screening process. After the publication of the protocol, a few changes were reported and submitted on September 8, 2023. These included the exclusion of pregnant and lactating women from the study population, and the inclusion of studies with a quasi-experimental design to account for potentially important articles. The literature search was performed on August 17, 2021, and updated on September 30, 2023.

## Data sources and search strategy

We considered five databases for our search—MEDLINE (via the PubMed interface), Scopus, Embase (via the embase.com interface), Web of Science, and the Cochrane Library (Cochrane Central Register of Controlled Trials). The search strategy for MEDLINE was first developed, and then translated for the other databases. The themes included in the search process were HIV/AIDS, nutritional status, adherence to ART, response to ART, nutritional interventions, adults, and LMICs. The detailed search protocols for the five databases are illustrated in the supplementary materials (see S1 Table–S5 Table).

## Inclusion and exclusion criteria

For the systematic review and meta-analyses, we included studies for further examination if 1) the study population consisted of HIV-positive adults (at least 18 years old) undergoing ART, 2) the study design was randomized controlled trial (RCT), controlled trial, quasi-experimental trial, longitudinal trial, or repeated cross-sectional evaluation, 3) they were conducted in LMICs, as classified by the World Bank [26], and 4) they assessed at least one of the following outcomes: nutritional status, immunological status, adherence to ART, food security, hemoglobin concentration, and micronutrient status. We did not consider any restrictions based on the type or duration of the nutritional intervention, the language, or the year of publication of the study. However, studies focusing exclusively on groups requiring additional nutritional care and non-ART medical treatment for co-morbidities were excluded. These include children under 18 years of age, pregnant or lactating women enrolled in the Prevention of Mother-to-Child Transmission program (for these programs include as well nutritional support), or individuals with comorbidities (e.g., tuberculosis). Studies that encompassed both adults and children, provided that subgroup data for adults were available, were included.

## Search selection

The records identified from the databases using the above-mentioned search strategy were exported to EndNote X20 (www.endnote.com), which identified and excluded duplicates after checking for true duplicates. Rayyan (https://rayyan.ai), a tool for systematic literature reviews, was used to screen the articles based on their title and abstract, and then with regard to their full text. The first-round screening of the records was conducted in April 2021 by two reviewers (AN and SA), who independently selected the relevant records for further analysis in accordance with the above-mentioned eligibility criteria. Disagreements between the two reviewers were resolved through consensus-based discussions. Later, the updated records were screened in September 2023 by one reviewer (AN).

## Outcomes of the studies

In the context of HIV/AIDS, nutritional intervention aims to improve the nutritional status and immune function of PLWHAs, limit their disease complications, and improve their quality of life and survival. Therefore, the main outcomes of the nutritional interventions considered in this review were differences between the baseline and endline (after the intervention) of nutritional status indicators, including weight, BMI, body fat mass, lean mass, and mid-upper arm circumference (MUAC), as well as immunological markers, such as CD4, viral load, and adherence to ART. Additional outcomes that were accounted for included hemoglobin, micronutrient concentration, and food security.

## Data extraction

The data extraction format used for the review was based on the Cochrane Data Collection Form for Intervention Reviews, which was modified according to the study question. AN and SA developed and pre-tested the data extraction form, following which they carried out data extraction between July and October 2023. Both reviewers independently assessed the eligibility of the papers by reading the entire text to ensure that they met the inclusion criteria. Disagreements about the inclusion or exclusion of an article were resolved through discussion. The interventions considered in this review included any nutritional intervention whose effects were compared to those pertaining to a control group of PLWHA that received no nutritional intervention. Notably, both the intervention and control groups should have received behavior change or standard counseling along with ART. This design enabled the examination of the sole effect of nutrition intervention on PLWHA. The data extraction form included information on the study design, inclusion and exclusion criteria, randomization, participants (age, sex), number of participants, study dropout, setting (country), intervention details (duration and frequency of intervention, duration of follow-up), and outcome measures. Furthermore, the articles selected for the systematic review were checked for their suitability for inclusion in the meta-analysis.

## Assessment of risk of bias

Next, two independent reviewers (AN and BTG) assessed the methodological quality of each study included in the review to evaluate the risk of bias (RoB) using the Cochrane risk of bias assessment tool (RoB 2) (https://sites.google.com/site/riskofbiastool/welcome/rob-2-0-tool). The following criteria were assessed: participant selection; blinding of participants, personnel, and outcome assessors; incomplete data; selective assessments of reports; withdrawals; and deviations from the protocol. The RoB assessment was conducted in October 2023.

## Statistical analysis

Statistical analysis was conducted using Review Manager (Review Manager 5.4). Descriptive statistics of the baseline characteristics were presented as mean and standard deviation (mean ± SD) or median with interquartile range (IQR) or 95% confidence interval (95% CI) for continuous variables, and as proportions for categorical variables.

The primary results were analyzed in terms of the type of outcome. Outcomes featuring continuous data were presented as pooled standard mean difference (SMD), calculated using the random effects model, with a 95% CI. Meanwhile, the categorical outcomes were calculated as relative risk estimates with a 95% CI. In cases where the SMD was not provided, the mean (SD) was estimated from the sample size and median (IQR) [27]. In the absence of SD, 95% CI and sample size were used to calculate it [28]. Furthermore, individual studies featuring multiple interventions were either presented and analyzed separately because of the difference in interventions (e.g., food basket vs. cash) or combined using the RevMan calculator, especially in the case of studies involving a single control group whose difference from the intervention group for the analyzed outcome was not significant. If the difference between the effects of these groups was significant, they were analyzed separately according to the Cochrane Review Manual [28].

In the case of the selected trials for which meta-analysis could not be performed due to insufficient data, a qualitative assessment was conducted. The study participants were considered the unit of analysis in randomized controlled, controlled, or quasi-experimental trials and repeated cross-sectional evaluations. Furthermore, in the case of cluster RCTs, clustering was accounted for in the analysis. Based on the recommendations of the Cochrane Manual,

the data analysis process in this research focused on effect size, direction of effect, and similarity between different studies. Moreover, the heterogeneity of the chosen studies was verified using Forest plot that incorporated Chi-square test ($\chi2$ test) and $I^2$ statistic to describe the percentage of variability in the effect estimates resulting from heterogeneity.

## Results

The search conducted across five databases was completed on August 2021 and was subsequently updated on September 2023 to identify newly published studies. Although the first literature search resulted in 6795 records, 3652 were excluded during the deduplication process using EndNote. An additional 68 records were identified from other sources. In the second round of search, 1592 new records were identified. Subsequently, the titles and abstracts of a total of 4735 records were screened (Fig 1). Among these, 4585 records were excluded during the title–abstract screening using Rayyan. The remaining 193 records that met our initial inclusion criteria were retained for full-text review. The full-text review resulted in the exclusion of 171 records for the following reasons: the PLWHA were not on ART, inadequate study

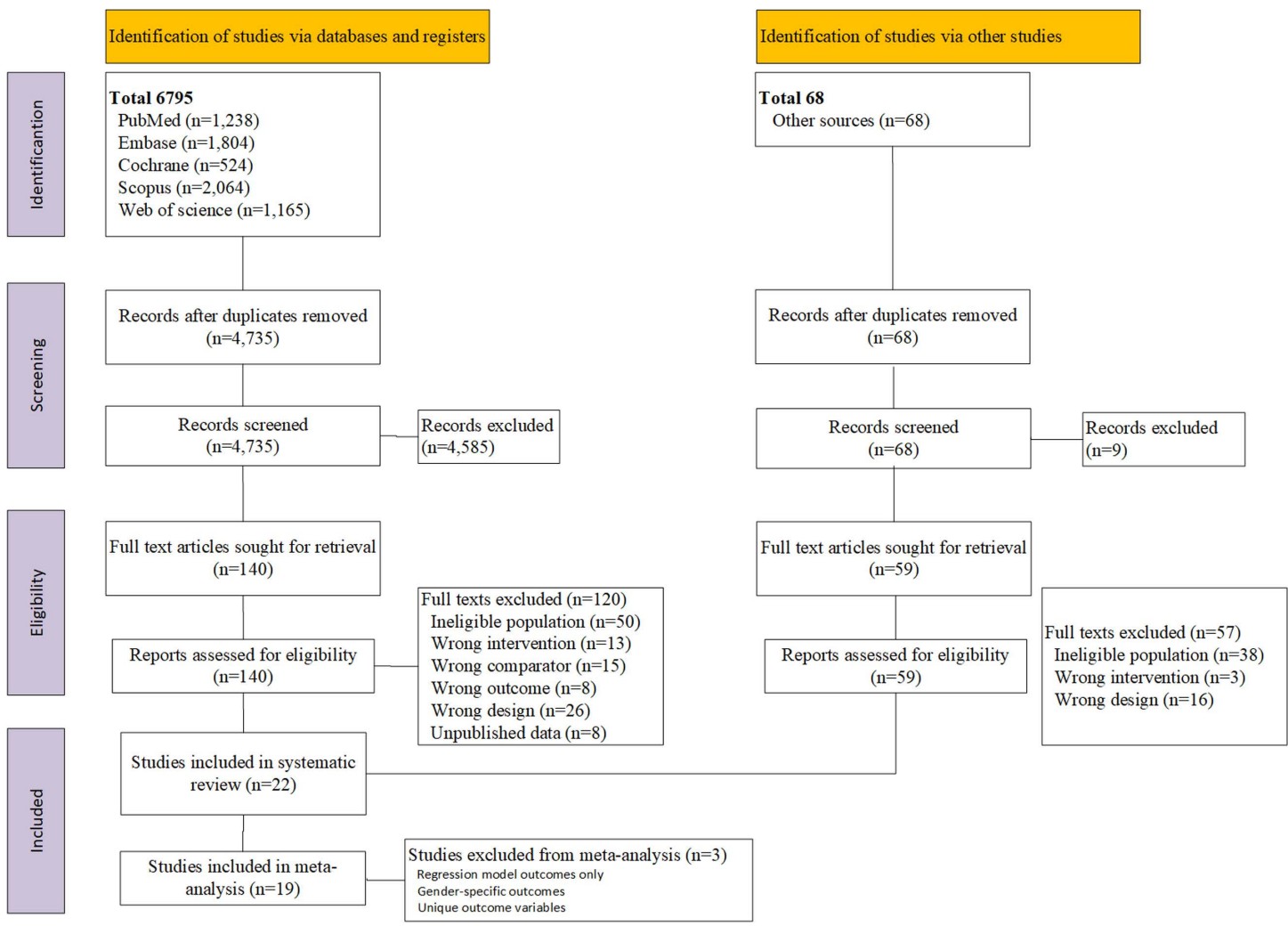

**Fig 1. PRISMA study flow diagram for the systematic review [ 29].**

design (village- or community-based intervention, program evaluation studies, or inappropriate study protocols), absence of a control group against which the nutritional intervention could be compared, studies dealing with only behavior change or nutritional counseling, studies with no published data, conference proceedings, registered studies that did not offer any outcomes, trials that failed to measure adequate outcomes, trials assessing the sustained long-term effects of an intervention, trials with ineligible study populations (hospitalized patients, patients under 18 years of age, patients with chronic comorbidities or co-infections, pregnant and lactating women, PLWHA with addiction, simultaneous involvement of children and adults, and combined outcomes), and trials conducted in inadequate settings (high-income countries). Additionally, studies targeting well-nourished PLWHA with the objective of reducing their oxidative stress and/or improving their metabolic syndrome markers, such as dyslipidemia, were excluded. Details of the reasons for excluding these studies are reported in the supplementary material (see S6 Table). Finally, 22 studies were included in the systematic review, and 19 were included in the meta-analysis.

## Description of the included studies

Our review examined 22 studies that reported on the outcomes attained from 17 trials involving 5464 participants. Data were extracted between September and November 2023 by AN and SA. The sample comprised 13 RCTs [30–45], two cluster RCTs [41,46], one cross-over RCT [47], and two quasi-experimental studies [48,49]. Notably, one of the selected studies conducted in Malawi comprised two cohorts, with the second one serving as a retrospective non supplemented control [50]. All studies were conducted between 2007 and 2021. The majority of the studies were conducted in Africa (Malawi [50], Tanzania [30,31], Ethiopia [32,34,35], Nigeria [33,37,43–45], Uganda [36], South Africa [38,40], the Democratic Republic of the Congo [39], Zambia [48], and Burkina Faso [49]), while three were based in Asia (Vietnam [47] and India [41,42,51]) and one in South America (Honduras [46]). The studies were conducted in 12 low- and middle-income countries (LMICs), including five low-income countries (Burkina Faso, Malawi, Ethiopia, the Democratic Republic of the Congo, and Uganda), six lower-middle-income countries (Tanzania, Nigeria, Zambia, Vietnam, India, and Honduras), and one upper-middle-income country (South Africa). All the studies were conducted in a hospital setting on PLWHA of both sexes aged 18 years or older, except for the studies conducted in India, in which all participants were women [41,42,51]. The duration of the interventions ranged from a minimum of 4 weeks [47] to a maximum of 18 months of supplementation [36]. Table 1 and Table 2 summarize the characteristics and data extracted from each of the included studies.

The selected studies had been conducted on participants with varied baseline nutritional statuses. Most participants had a normal weight with regard to their BMI range (18.5–24.9 kg/m²). However, the participants of the studies conducted in Malawi [50] and Vietnam [47] were largely emaciated, with their mean BMI at admission being < 18.5 kg/m². Furthermore, the selected studies included PLWHA who had recently started ART [32,34,35,37,40,48,50] or were already receiving ART [30,31,33,36,38,39,41–44,46,47,49]. The nutritional interventions administered in these studies included lipid-based nutrient supplements [32,34,35,47,50], corn soy blend (CSB) [50], food basket [30,31,46,48], prepared meals rich in proteins (boxes of meatballs and spaghetti with tomato sauce [38], porridge nutritional supplement [40], lentil- and pea-based meals [41,42], and optimized meals composed of soybean, millet, moringa, and carrot) [33], food vouchers [30,31], micronutrient supplements [36,37,45], *Moringa oleifera* lam. leaves in the form of powder [39,43,44] spirulina [49], and carrot-ginger extracts [45].

The key outcome measured by the selected studies was the nutritional status of the PLWHA, which pertained to estimations of their weight [31,35,36,38,40,41,43,48,50], BMI

**Table 1. Characteristics of the studies included in the systematic review.**

| Authors Year (Country) | Study design | Number of participants, age, and intervention duration | Intervention/control groups | Frequency of administration | Baseline characteristics | Primary outcomes | Included in meta-analysis |
|---|---|---|---|---|---|---|---|
| Brown et al. [47] 2015 (Vietnam) | Crossover control trial | N = 120; 34 (±7.0) years; 4 weeks | **Intervention 1:** Local RUTF<br>**Intervention 2:** Imported RUTF<br>**Control:** No supplementary food | Weekly | All on ART<br>BMI < 20 kg/m²<br>CD4 217.6 (±170.8) cells/μL | Nutritional status: Weight, BMI, MUAC | Yes |
| Van Oosterhout [50] 2010 (Malawi) | Two cohorts—one intervention group and one retrospective control group | N = 593; 18 years or older; 14 weeks | **Intervention 1:** Ready-to-use fortified spread (45% daily estimated average energy requirement)<br>**Intervention 2:** Corn/soy blended flour (45% daily estimated average energy requirement)<br>**Control:** No supplementary food | Monthly | Initiating ART<br>CD4 132.2 (±144.1) cells/μL<br>BMI < 18.5 kg/m² | Nutritional status: Weight, BMI<br>Severe clinical events<br>Lost to follow-up<br>Mortality rate<br>Adherence to ART (stopped) | Yes |
| McCoy et al. [30] 2017 (Tanzania) | RCT | N = 805 (777 on using a modified intention to treat protocol [31]); median age: 35 years; 12 months | **Intervention 1:** Nutrition assessment and counseling (NAC) + Cash transfers<br>**Intervention 2:** NAC and food basket<br>**Control:** Only NAC | Monthly | On ART (0–90 days)<br>Median CD4: 200 cells/μL | ART adherence (medication possession ratio)<br>Retention in care (lost to follow-up) | Yes |
| Fahey et al. [31] 2019 (Tanzania) | | | | | Median BMI: 21.0 kg/m² | Nutritional status: Weight, BMI<br>Food security<br>Livelihood-generating activities | Yes |
| Amlogu et al. [33] 2016 (Nigeria) | RCT | N = 50; 43.3 (±9.7) years; 6 months | **Intervention:** Optimized meal (354.92 kcal/d; soya bean, millet, moringa, and carrot)<br>**Control:** No supplementary food | Quarterly | On ART<br>MUAC 30.3 (±0.7) cm<br>CD4 385.2 (±73.2) cells/μL | Nutritional status: MUAC<br>Immunological status: CD4 | Yes |
| Gambo et al. [43] 2021 (Nigeria) | RCT | N = 177; ≥18 years; 6 months | **Intervention:** *Moringa oleifera* leaf powder (15 g/d)<br>**Control:** Placebo | Monthly | On ART (> 3 months)<br>BMI 24.3 (±4.3) kg/m² | Nutritional status: Weight, BMI<br>Immunological status: CD4, viral load | Yes |
| Tesfaye et al. [34] 2016 (Ethiopia) | RCT | N = 282; 32.8 (±9.1) years; 3 months | **Intervention 1:** LNS 200 g (with milk whey)<br>**Intervention 2:** LNS 200 g (with soya)<br>**Control:** Delayed LNS | Monthly | Initiating ART<br>BMI 19.9 (±2.2) kg/m² | Quality of life in people living with HIV<br>Food security | No |
| Olsen et al. [35] 2014 (Ethiopia) | | | | | CD4 187.7 (±103.3) cells/μL | Nutritional status: Weight, BMI, body composition (lean mass)<br>Immunological status: viral load, CD4<br>Grip strength<br>Physical activity | Yes |
| Yilma et al. [32] 2016 (Ethiopia) | | | | | | Body composition (Fat mass)<br>Serum 25(OH)D | Yes |

*(Continued)*

**Table 1.** (Continued)

| Authors Year (Country) | Study design | Number of participants, age, and intervention duration | Intervention/control groups | Frequency of administration | Baseline characteristics | Primary outcomes | Included in meta-analysis |
|---|---|---|---|---|---|---|---|
| Guwatudde et al. [36] 2015 (Uganda) | RCT | N = 400; 35.8 (±8.9) years; 18 months | **Intervention:** Multivitamins (1 RDA) **Control:** Placebo | Monthly | On ART (0–6 months on HAART) CD4 140.7 (±410.7) cells/µL BMI < 18.5kg/m² (7.8%), BMI > 25.0 kg/m² (27.0%) | Nutritional status: Weight, hemoglobin concentration Immunological status: CD4 Disease progression event Occurrence of an adverse event | Yes |
| Opara et al. [37] 2007 (Nigeria) | RCT | N = 290; ≥18 years; 4 months | **Intervention:** Nutritional counseling + MNP **Control:** No intervention | Every 2 weeks | Initiating ART | Biochemical indicators: Retinol, vitamin C, albumin, packed cell volume | No |
| Bhargava et al. [38] 2018 (South Africa)[1] | RCT | 643; 37.4 (±8.8); 12 months | **Intervention:** Adherence support for timely intake of ART from peer counselors + two 400-g cans of meat balls and spaghetti in tomato sauce per week **Control:** Adherence support for timely intake of ART from peer counselors **Control (negative):** Control ART | Two weekly counselling sessions and one weekly food supplement provision | On ART BMI 24.3 (±6.8) kg/m² CD4 239.6 (±148.2) cells/µL | Nutritional status: Weight, BMI Immunological status: Viral load, CD4 Food security | Yes |
| Tshingani et al. [39] 2017 (the Democratic Republic of the Congo) | RCT | N = 60; ≥18 years; 6 months | **Intervention:** Daily 30g *Moringa oleifera* lam. leaf powder + Nutritional counseling **Control:** Nutritional counseling | Monthly | On ART (~14 months) BMI 21.6 (±4.0) kg/m² CD4 445.9 (±203.9) cells/µL | Nutritional status: BMI, hemoglobin concentration Immunological status: Viral load, CD4 | Yes |
| Evans et al. [40] 2013 (South Africa) | RCT | N = 38; 35.5 (±38.3) years; 6 months | **Intervention:** FutureLife porridge® nutritional supplement (388 kcal/day) **Control:** No supplementary food | Monthly | Initiating ART BMI 19.8 (±12.3) kg/m² CD4 83.5 (±322.9) cells/µL | Nutritional status: Weight, BMI, body composition, hemoglobin concentration Immunological status: Viral load, CD4 Micronutrients: Calcium, selenium, magnesium, iron, ferritin | Yes |
| Cantrell et al. [48] 2008 (Zambia) | Quasi-experimental | N = 636; 36.5 (±8.5) years; 6 months initially, prolonged to 12 months if the household remains food insecure | **Intervention:** Food basket **Control:** No intervention | Monthly | Food insecure BMI 20.4 (±3.5) kg/m² CD4 131.1 (±107.2) cells/µL | Nutritional status: Weight Immunological status: CD4 Adherence to ART (medication possession) | Yes |

*(Continued)*

Table 1. (Continued)

| Authors Year (Country) | Study design | Number of participants, age, and intervention duration | Intervention/control groups | Frequency of administration | Baseline characteristics | Primary outcomes | Included in meta-analysis |
|---|---|---|---|---|---|---|---|
| Nyamathi et al. [41] 2018 (India) | Cluster RCT | N = 600; 34.3 (±7.0) years; 6 months | **Intervention 1:** ASHA + Nutrition education<br>**Intervention 2:** ASHA + Nutritional supplements (food baskets of high-protein dals, lentils, black grams, and pigeon peas)<br>**Intervention 3:** ASHA + Nutrition education + Nutritional supplements<br>**Control:** ASHA | Weekly | On ART (>3 months) BMI 20.1 (±4.2) kg/m² CD4 447.4 (±273.6) cells/ μL | Nutritional status: Weight, BMI Immunological status: CD4 | Yes |
| Carpenter et al. [42] 2021 (India) | | | | | | Nutritional status: Body composition (fat mass and lean mass) | Yes |
| Nyamathi et al. [51] 2019 (India) | | | | | | Hemoglobin concentration | Yes |
| Martinez et al. [46] 2014 (Honduras) | Cluster RCT | N = 400; 40 (±0.49) years; 12 months | **Intervention:** Food basket + Nutrition education<br>**Control:** Nutrition education | Monthly | On ART (average 3.7 years) BMI 24 (±4.5) kg/m² CD4 293 (±175.6) cells/ μL | Adherence: Missed clinic appointments, delayed prescription refills, self-reported missed doses of ART | No |
| Ouedraogo et al. [49] 2013 (Burkina Faso) | Quasi-experimental | N = 100; 37.8 (±8.7) years; 9 months | **Intervention:** Spirulina (10 g daily)<br>**Control:** No supplement | Monthly for the first 3 months, then quarterly | On ART BMI 19.3 (±3.3) kg/m² | Nutritional status: BMI, MUAC, hemoglobin concentration Adherence to ART Death | Yes |
| Aprioku et al.[44] 2022 (Nigeria) | RCT | N = 104; 21–70 years; 3 months | **Intervention:** Moringa (200 mg daily)<br>**Control:** No supplement | At 1 month and at 3 months | On HAART CD4 427 (±248) cells/ μL | Nutritional status: Hemoglobin concentration Immunological status: CD4 | Yes |
| Joshua et al. [45] 2021 (Nigeria) | RCT | N = 90; 18–>50 years; 3 months | **Intervention 1:** Micronutrient supplements (vitamins A, C, and E, selenium and zinc), one tablet daily<br>**Intervention 2:** Carrot-ginger blend (75:25), 2 sachets daily<br>**Control:** No supplement | Monthly | ART naive, just started BMI 22.4 (±1.8) kg/m² CD4 374 (±16) cells/ μL | Nutritional status: BMI Immunological status: CD4 | Yes |

ART, antiretroviral treatment; ASHA, accredited social health activist; BMI, body mass index; HAART, highly active antiretroviral therapy; LNS, lipid-based nutrient supplement; MUAC, mid-upper arm circumference; RCT, randomized control trial; RUTF, ready-to-use therapeutic food; RDA, recommended dietary allowance.

†Descriptive results reported for two out of the three groups (intervention and control).

**Table 2. Data extracted from the studies included in the systematic review.**

| Authors | Measured outcome | Reported statistics | Control | Intervention 1 | Intervention 2 |
|---|---|---|---|---|---|
| Brown et al. [47] | Change in BMI (kg/m²) | Mean ± SD (n) | 0.10 ± 0.27 (31) | 0.40 ± 0.68 (59) | |
| | Change in MUAC (cm) | Mean ± SD (n) | 0.41 ± 3.23 (29) | −1.89 ± 19.17 (54) | |
| Van Oosterhout [50] | Change in weight (kg) | Mean ± SD (n) | 3.3 ± 4.7 (104) | 5.6 ± 4.8 (244) | 4.4 ± 4.3 (245) |
| | Change in BMI (kg/m²) | Mean ± SD (n) | 1.2 ± 1.8 (104) | 2.2 ± 1.9 (244) | 1.7 ± 1.6 (245) |
| | Adherence to ART (stopped) | N (%) | 9 (9) | 1 (0.4) | 3 (1) |
| McCoy et al. [30] | ART adherence (MPR) | % (n) | 85.4 (112) | 95.1 (346) | 92.9 (342) |
| | Retention in care (lost to follow-up) | % (n) | 10.9 (112) | 0.9 (346) | 1.5 (342) |
| Fahey et al. [31] | Change in weight (kg) | Mean difference (95% CI) (n) | 2.31 (1.39–3.23) (111) | 2.40 (1.90–2.91) (334) | 2.84 (2.30–3.37) (332) |
| | Change in BMI (kg/m²) | Mean difference (95% CI) (n) | 0.91 (0.57–1.25) (111) | 0.93 (0.73–1.14) (334) | 1.11 (0.89–1.32) (332) |
| | Change in household food insecurity access score | Mean difference (95% CI) (n) | −4.10 (−6.54−−1.66) (111) | −6.03 (−7.18−−4.88) (334) | −5.58 (−6.70−−4.45) (332) |
| Amlogu et al. [33] | Endline MUAC (cm) | Mean ± SD (n) | 29.01 ± 0.69 (25) | 31.78 ± 0.69 (25) | |
| | Endline CD4 (cells/mm³) | Mean ± SD (n) | 306.07 ± 75.00 (25) | 469.26 ± 75.00 (25) | |
| Gambo et al. [43] | Endline weight (kg) | Mean ± SD (n) | 63.16 ± 13.49 (88) | 64.71 ± 15.07 (89) | |
| | Endline BMI (kg/m²) | Mean ± SD (n) | 24.19 ± 4.09 (88) | 25.16 ± 4.93 (89) | |
| | Endline CD4 (cells/mm³) | Mean ± SD (n) | 373.44 ± 157.31 (88) | 425.75 ± 153.76 (89) | |
| Tesfaye et al. [34] | Food security prevalence | % with no food insecurity | 14.0 | 25.5 | |
| | | % with mild food insecurity | 10.0 | 23.1 | |
| | | % with moderate food insecurity | 49.2 | 26.2 | |
| | | % with severe food insecurity | 25.6 | 25.6 | |
| Olsen et al. [35] | Change in weight at 3 months (kg) | Unadjusted mean (95% CI) (n) | 0.75 (0–1.5) (83) | 2.67 (−1.92–3.42) (79) | 2.83 (−2.25–3.41) (87) |
| | Change in fat-free mass at 3 months (kg) | Unadjusted mean (95% CI) (n) | 0.06 (−0.43–0.55) (83) | 0.85 (0.16–1.53) (79) | 0.97 (0.29–1.64) (87) |
| | Change in CD4 at 3 months (cells/μL) | Unadjusted mean (95% CI) (n) | 85.7 (66.7–104.7) (83) | 114 (95–133) (79) | 95.2 (76–114) (87) |
| | Change in viral load at 3 months (log 1+(copies/mL)) | Unadjusted mean (95% CI) (n) | −3.75 (−4.08−−3.42) (83) | −3.75 (−4.08−−3.42) (79) | −3.75 (−3.67−−3.17) (87) |
| Yilma et al. [32] | Change in fat mass at 3 months (kg) | Median (IQR) (n) | 0.52 (−0.87–1.75) (93) | 1.73 (0.18–3.6) (189) | |
| Guwatudde et al. [36] | Change in weight (kg) | Mean ± SD (n) | 3.3 ± 5.8 (187) | 3.9 ± 6.2 (181) | |
| | Change in Hb concentration (g/dL) | Mean ± SD (n) | 0.91 ± 1.7 (186) | 1.01 ± 1.5 (181) | |
| | Change in CD4 (cells/μL) | Mean ± SD (n) | 147 ± 130 (186) | 141 ± 160 (181) | |
| Opara et al. [37] | Mean change in packed cell volume (%) | Mean (Male) | −2.0 | 5.5 | |
| | | Mean ± SD (Female) | −2.8 ± 1.5 | 5.3 ± 0.5 | |
| | Mean change in serum retinol (μg/L) | Mean ± SD (Male) | −2.2 ± 3.8 | 12.1 ± 1.1 | |
| | | Mean ± SD (Female) | −2.8 ± 0.75 | 13.9 ± 1.6 | |
| | Mean change in serum vitamin C (mg/L) | Mean ± SD (Male) | −0.5 ± 0.03 | 0.95 ± 0.39 | |
| | | Mean ± SD (Female) | 0.07 ± 0.04 | 0.93 ± 0.5 | |
| Bhargava et al. [38] | Endline weight (kg) | Mean ± SD (n) | 65.0 ± 14.0 (135) | 63.9 ± 16.2 (127) | |
| | Endline BMI (kg/m²) | Mean ± SD (n) | 25.4 ± 6.0 (135) | 25.5 ± 7.1 (127) | |
| | Endline CD4 (cells/mm³) | Mean ± SD (n) | 458.9 ± 235.7 (135) | 426.2 ± 197.8 (127) | |
| | Viral load (ln HIV RNA, copies/mL) | Mean ± SD (n) | 4.29 ± 1.97 (135) | 4.82 ± 2.09 (127) | |
| | Food insecurity index | Mean ± SD (n) | 7.31 ± 5.38 (135) | 6.59 ± 5.42 (127) | |

*(Continued)*

**Table 2.** (Continued)

| Authors | Measured outcome | Reported statistics | Control | Intervention 1 | Intervention 2 |
|---------|-----------------|---------------------|---------|----------------|----------------|
| Tshingani et al. [39] | BMI at 6 months (kg/m²) | Mean ± SD (n) | 22.4 ± 3.7 (29) | 24.4 ± 4.0 (29) | |
| | Change in CD4 (cells/mm³) | Mean ± SD (n) | 63.9 ± 107.3 (29) | 51.3 ± 100.0 (29) | |
| Evans et al. [40] | Endline weight (kg) | Median (IQR) (n) | 54.0 (50.2–61.3) (15) | 59.6 (48.0–65.4) (11) | |
| | Endline BMI (kg/m²) | Median (IQR) (n) | 20.0 (18.5–23.7) (15) | 22.1 (19.9–24.3) (11) | |
| | Endline fat mass (kg) | Median (IQR) (n) | 11.9 (9.5–20.6) (15) | 12.8 (11.4–22.9) (11) | |
| | Endline fat-free mass (kg) | Median (IQR) (n) | 39.8 (35.5–43.1) (15) | 42.5 (35.8–50.9) (11) | |
| | Endline Hb concentration (g/dL) | Median (IQR) (n) | 13.6 (12.8–14.7) (15) | 12.5 (11.1–13.6) (11) | |
| | Endline CD4 (cells/mm³) | Median (IQR) (n) | 233 (152–359) (15) | 167 (154–293) (11) | |
| | Endline iron concentration (μmol/L) | Median (IQR) (n) | 13.2 (12.1–18.9) (15) | 11.3 (6.7–11.8) (11) | |
| | Adherence to ART | Events/total | 14/15 | 11/11 | |
| Cantrell et al. [48] | Change in weight at 12 months (kg) | Mean (95% CI) (n) | 5.4 (4.0–6.8) (113) | 6.3 (5.5–7.2) (302) | |
| | Change in CD4 at 12 months (cells/mm³) | Mean (95% CI) (n) | 182 (160–204) (105) | 180 (147–214) (241) | |
| | Adherence to ART (medication possession) | Events/total (n) | 79/166 | 258/366 | |
| Nyamathi et al. [41] *Control vs. Nutrition supplements* | Change in weight (kg) | Mean ± SD (n) | 2.17 ± 1.80 (150) | 5.46 ± 1.33 (150) | |
| | Change in BMI (kg/m²) | Mean ± SD (n) | 0.95 ± 0.82 (150) | 2.38 ± 0.60 (150) | |
| | Change in CD4 (cells/mm³) | Mean ± SD (n) | 343.97 ± 106.94 (150) | 469.66 ± 116.00 (150) | |
| *Nutrition education vs. Nutrition education and supplements* | Change in weight (kg) | Mean ± SD (n) | 2.91 ± 1.19 (150) | 6.24 ± 1.84 (150) | |
| | Change in BMI (kg/m²) | Mean ± SD (n) | 1.28 ± 0.53 (150) | 2.72 ± 0.84 (150) | |
| | Change in CD4 (cells/mm³) | Mean ± SD (n) | 356.15 ± 80.69 (150) | 530.82 ± 128.56 (150) | |
| Carpenter et al. [42] *Control vs. Nutrition supplements* | Fat mass at 6 months (kg) | Mean ± SD (n) | 12.19 ± 5.09 (150) | 14.79 ± 6.89 (150) | |
| | Fat-free mass at 6 months (kg) | Mean ± SD (n) | 35.28 ± 6.24 (150) | 38.10 ± 7.16 (150) | |
| *Nutrition education vs. Nutrition education and supplements* | Fat mass at 6 months (kg) | Mean ± SD (n) | 13.55 ± 5.28 (150) | 13.91 ± 5.89 (150) | |
| | Fat-free mass at 6 months (kg) | Mean ± SD (n) | 35.78 ± 6.75 (150) | 38.17 ± 6.82 (150) | |
| Nyamathi et al. [51] *Control vs. Nutrition supplements* | Hb concentration at 6 months (g/L) | Mean ± SD (n) | 102.86 ± 2.86 (150) | 105.71 ± 2.86 (150) | |
| *Nutrition education vs. Nutrition education and supplements* | Hb concentration at 6 months (g/L) | Mean ± SD (n) | 104.29 ± 2.86 (150) | 108.57 ± 2.86 (150) | |
| Martinez et al. [46] | Missed clinic appointments at 12 months | % | 3.3 | 18.8 | |
| | Delayed prescription refills at 6 and 12 months | % | 2.7 | 22.6 | |
| | Self-reported missed ART doses at 6 and 12 months | % | 5.4 | 6.6 | |
| Ouedraogo et al. [49] | Endline BMI (kg/m²) | Mean ± SD (n) | 21.08 ± 2.73 (37) | 21.90 ± 3.56 (32) | |
| | Endline MUAC (cm) | Mean ± SD (n) | 25.86 ± 2.43 (37) | 27.44 ± 2.25 (32) | |
| Aprioku et al.[44] | Endline Hb concentration at 3 months (g/dL) | Mean ± SD (n) | 11.84 ± 2.69 (104) | 13.15 ± 1.33 (104) | |
| | Endline CD4 at 3 months (cells/μL) | Mean ± SD (n) | 482.73 ± 187.53 (104) | 582.79 ± 271.66 (104) | |
| Joshua et al. [45] | Endline BMI at 90 days (cells/μL) | Mean ± SD (n) | 22.96 ± 0.95 (30) | 26.65 ± 0.68 (30) | 25.17 ± 0.64 (30) |
| | Endline CD4 at 90 days (cells/μL) | Mean ± SD (n) | 380.47 ± 11.02 (30) | 477.23 ± 8.29 (30) | 401.86 ± 9.03 (30) |
| | Endline β-carotene concentration (μg/dL) | Mean ± SD (n) | 32.09 ± 0.19 (30) | 38.02 ± 0.62 (30) | 41.89 ± 0.61 (30) |
| | Endline zinc concentration (μg/dL) | Mean ± SD (n) | 61.31 ± 0.13 (30) | 70.15 ± 0.27 (30) | 65.21 ± 0.28 (30) |

ART, antiretroviral treatment; BMI, body mass index; CI, confidence interval; Hb, hemoglobin; IQR, interquartile range; MUAC, mid-upper arm circumference.

[31,38–41,43,45,47,49,50], and body composition, including fat mass [32,40,42], fat-free mass [35,40,42], and MUAC [33,47,49]. The second outcome examined by the studies was immunological response, summarized as CD4 count [33,35,36,38–41,43–45,48] and viral load [38,52]. A few studies also examined adherence to ART, assessed during routine follow-up visits [30,48]. The secondary outcomes include hemoglobin concentration as a marker of anemia, which was reported in two studies [36,40]. Other studies that involved food-based interventions also measured food insecurity [31,38]. Furthermore, a few single studies reported on the effects of micronutrients, such as vitamins D and A, zinc, and iron [37,40,45].

## Risk of bias in the included studies

The 22 studies considered in this study, which reported on a total of 17 trials, provided evidence of varying strengths (Table 3 and S1 Fig). Seven trials, whose outcomes were reported in 11 studies, were ranked as having a low RoB. Meanwhile, two trials reported in three studies raised some concerns about RoB pertaining to deviations from the main protocol. For instance, studies comparing the effects of cash against those of food baskets, which included moderately malnourished participants, did not analyze the outcomes with regard to BMI [30,31]. Furthermore, seven studies exhibited high RoB, largely owing to weak randomization procedure details. For instance, in a crossover RCT conducted in Vietnam, no randomization was implemented in the control and intervention groups [47]. Additionally, for the entire duration of the intervention, the outcomes of the intervention groups were considered to be that of one group featuring a larger sample size [47]. Moreover, three studies provided no details on their randomization procedures [33,37,38], while one study used an ad hoc control group for comparison [50]. The reporting of only a few selected outcomes was also evaluated as implying high RoB, such as estimating only MUAC and not weight [33], examining BMI and not weight [39,49], and reporting secondary analyses (effect by sex) but not primary analyses (group effect) [37]. Furthermore, Ouedraogo et al. did not report the details of the measurements they conducted [49]. In the study by Bhargava et al., some participants in the control group who should have received only nutrition counseling also received CSB [38].

## Effects of nutritional intervention on nutritional status

The effects of nutritional interventions on the nutritional status of PLWHA were assessed based on their weight, BMI, and body composition, including fat mass, fat-free mass, and MUAC.

**Effects of nutritional intervention on weight.** Changes in weight were reported in 10 studies that involved the provision of lipid-based nutrient supplements [35,47,50], CSB [50], food baskets [31,41,48], micronutrients [36], functional food [43], conditional cash [31], and prepared meals [38] to PLWHA. In four of the studies, a positive effect of the interventions on change in weight was observed in the intervention groups compared to the control groups [35,41,47,50]. Among these, Brown et al. reported a significant difference in percent weight gain between PLWHA who consumed ready-to-use therapeutic food (2.3%, n = 59) and those in the control group (0.5%, n = 31, p = 0.017) [47]. Meanwhile, Olsen et al. showed that daily supplementation of PLWHA (BMI > 17 kg/m²) with 200 g lipid-based nutrient supplements ($\sim$ 1100 Kcal) for 3 months significantly increased their weight by 2.75 (±3.07) kg, compared to the delayed intervention that registered a mean weight gain of 0.75 (±3.49) kg [35]. Furthermore, Van Oosterhout et al. conducted a 14-week study in Malawi and reported a significant weight gain in two intervention groups that received lipid-based nutrient supplements and CSB, respectively, compared to the results observed for the retrospective control group [50]. Notably, the participants in two of these studies were entirely

**Table 3. Risk of bias assessment using the Cochrane risk of bias assessment tool (RoB 2).**

**Studies that assessed the effect of nutritional interventions on change in weight as the outcome of interest**

| Author, reference | D1 | D2 | D3 | D4 | D5 | Overall |
|---|---|---|---|---|---|---|
| Olsen et al. [35] | Low | Low | Low | Low | Low | Low |
| Van Oosterhout et al. [50] | High | Low | Low | Low | Low | High |
| Cantrell et al. [48] | Some concern | Some concern | Low | Low | Low | Some concern |
| Fahey et al. [31] | Low | Some concern | Low | Low | Low | Some concern |
| Nyamathi et al. [41] | Low | Low | Low | Low | Low | Low |
| Guwatudde et al. [36] | Low | Low | Low | Low | Low | Low |
| Gambo et al. [43] | Low | Low | Low | Low | Low | Low |
| Bhargava et al. [38] | High | Low | Low | Low | Low | High |
| Evans et al. [40] | Low | Low | Low | Low | Low | Low |

**Studies that assessed the effect of nutritional interventions on change in BMI as the outcome of interest**

| Author, reference | D1 | D2 | D3 | D4 | D5 | Overall |
|---|---|---|---|---|---|---|
| Brown et al. [47] | High | Low | Low | Low | Low | High |
| Van Oosterhout et al. [50] | High | Low | Low | Low | Low | High |
| Fahey et al. [31] | Low | Some concern | Low | Low | Low | Some concern |
| Nyamathi et al. [41] | Low | Low | Low | Low | Low | Low |
| Joshua et al. [45] | Low | Low | Low | Low | Low | Low |
| Gambo et al. [43] | Low | Low | Low | Low | Low | Low |
| Ouedraogo et al. [49] | Low | Low | Low | High | High | High |
| Tshingani et al. [39] | Low | Low | Low | Low | High | High |
| Bhargava et al. [38] | High | Low | Low | Low | Low | High |
| Evans et al. [40] | Low | Low | Low | Low | Low | Low |

**Studies that assessed the effect of nutritional interventions on body fat mass as the outcome of interest**

| Author, reference | D1 | D2 | D3 | D4 | D5 | Overall |
|---|---|---|---|---|---|---|
| Yilma et al. [32] | Low | Low | Low | Low | Low | Low |
| Carpenter et al. [42] | Low | Low | Low | Low | Low | Low |
| Evans et al. [40] | Low | Low | Low | Low | Low | Low |

**Studies that assessed the effect on body fat-free mass as the outcome of interest**

| Author, reference | D1 | D2 | D3 | D4 | D5 | Overall |
|---|---|---|---|---|---|---|
| Olsen et al. [35] | Low | Low | Low | Low | Low | Low |
| Carpenter et al. [42] | Low | Low | Low | Low | Low | Low |
| Evans et al. [40] | Low | Low | Low | Low | Low | Low |

**Studies that assessed the effect on mid-upper arm circumference as the outcome of interest**

| Author, reference | D1 | D2 | D3 | D4 | D5 | Overall |
|---|---|---|---|---|---|---|
| Brown et al. [47] | High | Low | Low | Low | Low | High |
| Amlogu et al. [33] | High | Low | High | Low | High | High |
| Ouedraogo et al. [49] | Low | Low | Low | High | High | High |

**Studies that assessed the effect of nutritional interventions on hemoglobin concentrations as the outcome of interest**

| Author, reference | D1 | D2 | D3 | D4 | D5 | Overall |
|---|---|---|---|---|---|---|
| Guwatudde et al. [36] | Low | Low | Low | Low | Low | Low |
| Aprioku et al. [44] | Low | Low | Low | Low | Low | Low |
| Evans et al. [40] | Low | Low | Low | Low | Low | Low |
| Nyamathi et al. [51] | Low | Low | Low | Low | Low | Low |

**Studies that assessed the effect of nutritional interventions on micronutrient concentration as the outcome of interest**

| Author, reference | D1 | D2 | D3 | D4 | D5 | Overall |
|---|---|---|---|---|---|---|
| Opara et al. [37] | High | Low | Low | Low | High | High |
| Evans et al. [40] | Low | Low | Low | Low | Low | Low |
| Joshua et al. [45] | Low | Low | Low | Low | Low | Low |

*(Continued)*

**Table 3.** (Continued)

**Studies that assessed the effect of nutritional interventions on change in weight as the outcome of interest**

**Studies that assessed the effect on CD4 as the outcome of interest**

| Author, reference | D1 | D2 | D3 | D4 | D5 | Overall |
|---|---|---|---|---|---|---|
| Olsen et al. [35] | Low | Low | Low | Low | Low | Low |
| Cantrell et al. [48] | Some concern | Some concern | Low | Low | Low | Some concern |
| Nyamathi et al. [41] | Low | Low | Low | Low | Low | Low |
| Guwatudde et al. [36] | Low | Low | Low | Low | Low | Low |
| Joshua et al. [45] | Low | Low | Low | Low | Low | Low |
| Aprioku et al. [44] | Low | Low | Low | Low | Low | Low |
| Gambo et al. [43] | Low | Low | Low | Low | Low | Low |
| Tshingani et al. [39] | Low | Low | Low | Low | High | High |
| Amlogu et al. [33] | High | Low | High | Low | High | High |
| Bhargava et al. [38] | High | Low | Low | Low | Low | High |
| Evans et al. [40] | Low | Low | Low | Low | Low | Low |

**Studies that assessed the effect of nutritional interventions on viral load as the outcome of interest**

| Author, reference | D1 | D2 | D3 | D4 | D5 | Overall |
|---|---|---|---|---|---|---|
| Olsen et al. [35] | Low | Low | Low | Low | Low | Low |
| Bhargava et al. [38] | High | Low | Low | Low | Low | High |

**Studies that assessed the effect of nutritional interventions on adherence to ART as the outcome of interest**

| Author, reference | D1 | D2 | D3 | D4 | D5 | Overall |
|---|---|---|---|---|---|---|
| Cantrell et al. [48] | Some concern | Some concern | Low | Low | Low | Some concern |
| McCoy et al. [30] | Low | Some concern | Low | Low | Low | Some concern |
| Evans et al. [40] | Low | Low | Low | Low | Low | Low |
| Martinez et al. [46] | Low | Low | Low | Low | Low | Low |

**Studies that assessed the effect of nutritional interventions on food security as the outcome of interest**

| Author, reference | D1 | D2 | D3 | D4 | D5 | Overall |
|---|---|---|---|---|---|---|
| Fahey et al. [31] | Low | Some concern | Low | Low | Low | Some concern |
| Bhargava et al. [38] | High | Low | Low | Low | Low | High |
| Tesfaye et al. [34] | Low | Low | Low | Low | Low | Low |

D1: Randomization process; D2: Intervention deviations; D3: Missing outcome data; D4: Measurement of the outcome; D5: Selection of the reported result

[50] or mostly [47] malnourished PLWHA with a BMI < 18.5 kg/m². Moreover, Nyamathi et al. conducted a cluster RCT targeting women with a positive effect of the provision of food baskets composed of plant-based proteins over a period of 6 months on the participants' weight [41]. All other studies reported a non-significant effect of nutritional interventions on weight compared to their respective control groups.

Overall, nutritional supplementation improved the weight status of PLWHA compared to the control group in only three studies, among which one was ranked as having high RoB [50] while two were characterized by low RoB [35,41].

**Effects of nutritional intervention on change in weight: Meta-analysis.** Nine studies reporting on 11 interventions were included in the meta-analysis (Fig 2) [31,35,36,38,40,41,43,48,50]. The study by Brown et al. reported on change in weight, but did not provide details on the standard deviations or 95% CI, as a result of which it was excluded from the meta-analysis [47]. A total of 2331 and 1136 adults of both sexes received nutritional intervention and no or delayed intervention, respectively. Overall, nutritional interventions had no significant effect on change in weight among PLWHA (SMD 0.36; 95% CI: -0.09, 0.81; P = 0.12; test for heterogeneity: $I^2$ = 97%, P < 0.00001).

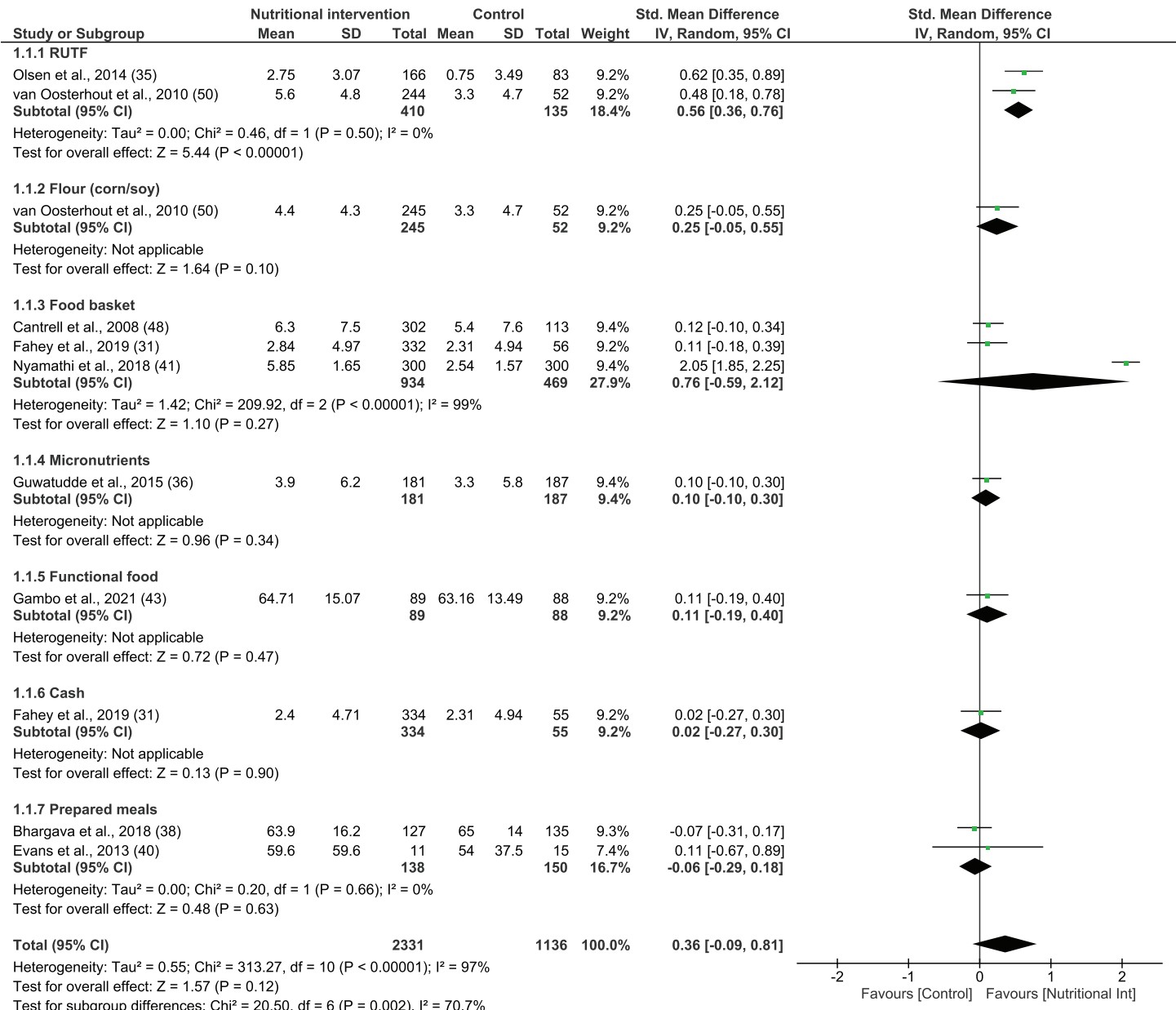

**Fig 2. Forest plot of SMD analysis results of the effect of nutritional interventions on weight gain among people living with HIV/AIDS.**

A subgroup analysis was performed based on the type of nutritional intervention, RoB, duration of intervention, and the income status of the countries. Notably, lipid-based nutrient supplements were the only interventions whose combined effect on changes in weight among PLWHA was found to be positive (2 studies; SMD 0.54; 95% CI: 0.36, 0.72; P < 0.00001; n = 597; test for heterogeneity: $I^2$ = 0%, P = 0.44). None of the other subgroup analyses exhibited any effect on weight compared to the control group, including food baskets (3 studies; SMD 0.76; 95% CI: -0.54, 2.06; P = 0.25; n = 1458; test for heterogeneity: $I^2$ = 99%, P < 0.00001) and prepared meals (2 studies; SMD -0.06, 95% CI: -0.29, 0.18; P = 0.63; n = 288; test for heterogeneity: $I^2$ = 0%, P = 0.66). CSB, micronutrients, functional food, and conditional cash transfer interventions did not yield any evidence related to weight gain (see S2 Fig). Among

the studies included in the meta-analysis, five achieved low RoB, two attained moderate RoB, and two had high RoB. Overall, the subgroup analysis, which was conducted based on the RoB, revealed no significant effect of the interventions on the weight of PLWHA across all RoB subgroups (see S3 Fig). Furthermore, while interventions lasting less than 4 months had a significant effect on the weight of PLWHA (SMD 0.46; 95% CI: 0.24, 0.67; P < 0.01; n = 842; test for heterogeneity: $I^2$ = 38%, P = 0.20), those lasting 4 to 6 months (SMD 0.61; 95% CI: -0.56, 1.78; P = 0.31; n = 1218; test for heterogeneity: $I^2$ = 99%, P < 0.01) and over 6 months (SMD 0.04; 95% CI: -0.08, 0.16; P = 0.51, n = 1407; test for heterogeneity: $I^2$ = 0%, P = 0.71) showed no such effect (see S4 Fig). Interventions conduced in low-income countries (SMD 0.35; 95% CI: 0.10, 0.60; P < 0.01; n = 1210; test for heterogeneity: $I^2$ = 71%, P = 0.01) showed significant weight gain among PLWHA. However, interventions in lower-middle-income (SMD 0.48; 95% CI: -0.42, 1.38; P = 0.29; n = 1969; test for heterogeneity: $I^2$ = 98%, P < 0.01) and upper-middle-income countries (SMD -0.06; 95% CI: -0.29, 0.18; P = 0.63; n = 288; test for heterogeneity: $I^2$ = 0%, P < 0.66) showed no significant effect (see S5 Fig).

**Effects of nutritional intervention on change in BMI.** In this research, 10 of the selected studies reported on the effect of 13 nutritional interventions on BMI [31,38–41,43,45,47,49,50]. The interventions involved the provision of lipid-based nutrient supplements [47,50], CSB [50], functional foods and micronutrients [45], and food baskets [41] to improve the participants' BMI compared to the control group. For instance, Brown et al. conducted a crossover RCT in which ready-to-use therapeutic foods were provided to PLWHA, the majority of whom were malnourished, for 4 weeks (89% PLWHA had BMI < 18.5 kg/$m^2$ in the intervention group vs. 82% in the control group) [47]. Six of the selected studies did not find any significant effect of nutritional interventions on the BMI of PLWHA [31,38–40,43,49]. Furthermore, two of the studies that identified an impact on BMI were ranked as having high RoB [47,50], while the other two exhibited low RoB [41,45].

**Effects of nutritional intervention on change in BMI: Meta-analysis.** Similar to the systematic review, the effectiveness of nutritional interventions on BMI was examined by considering 10 studies that reported on 13 different interventions [31,38–41,43,45,47,49,50]. The combined effect of the nutritional interventions on change in BMI of PLWHA compared to those who did not receive any such intervention was significant (13 interventions; SMD 0.80; 95% CI: 0.36, 1.24; P = 0.0004; n = 2442; test for heterogeneity: $I^2$ = 95%, P < 0.00001) (Fig 3).

A subgroup analysis was conducted based on the type of nutritional intervention, RoB, duration of intervention, and the income status of the countries. Regarding the interventions, no evidence was found on the effectiveness of CSB, micronutrients, or conditional cash. The effect on BMI was statistically significant only in the case of interventions that provided lipid-based nutrient supplements (SMD 0.53; 95% CI: 0.32, 0.74; P < 0.00001; n = 438; test for heterogeneity: $I^2$ = 0%, P = 0.95). Moreover, food baskets (SMD 1.03; 95% CI: -0.79, 2.85; P = 0.27; n = 743; test for heterogeneity: $I^2$ = 99%, P < 0.01), functional food (SMD 0.87; 95% CI: -0.00, 1.75; P = 0.05; n = 364; test for heterogeneity: $I^2$ = 93%, P < 0.01), and prepared meals (SMD 0.02; 95% CI: -0.21, 0.26; P = 0.83; n = 288; test for heterogeneity: $I^2$ = 0%, P = 0.80) had no effect on the BMI of PLWHA (see S6 Fig). Among the 10 studies considered for this analysis, four had low RoB, five exhibited high RoB, and one had moderate RoB. The studies with low RoB (SMD 1.96; 95% CI 0.65, 1.89; P < 0.003; n = 593; test for heterogeneity: $I^2$ = 97%, P < 0.01) and high RoB (SMD 0.32; 95% CI: 0.13, 0.52; P < 0.01; n = 1072; test for heterogeneity: $I^2$ = 45%, P = 0.11) showed a significant effect the nutritional interventions on the BMI of PLWHA (S7 Fig). Furthermore, while interventions that lasted less than 4 months had a significant effect on the BMI of PLWHA (SMD 1.58; 95% CI: 0.68, 2.48; P < 0.01; n = 773; test for heterogeneity: $I^2$ = 95%, P < 0.01), those that lasted at least 4 months did not exhibit any such effect (see S8 Fig). The interventions in low-income countries (SMD 0.40; 95%

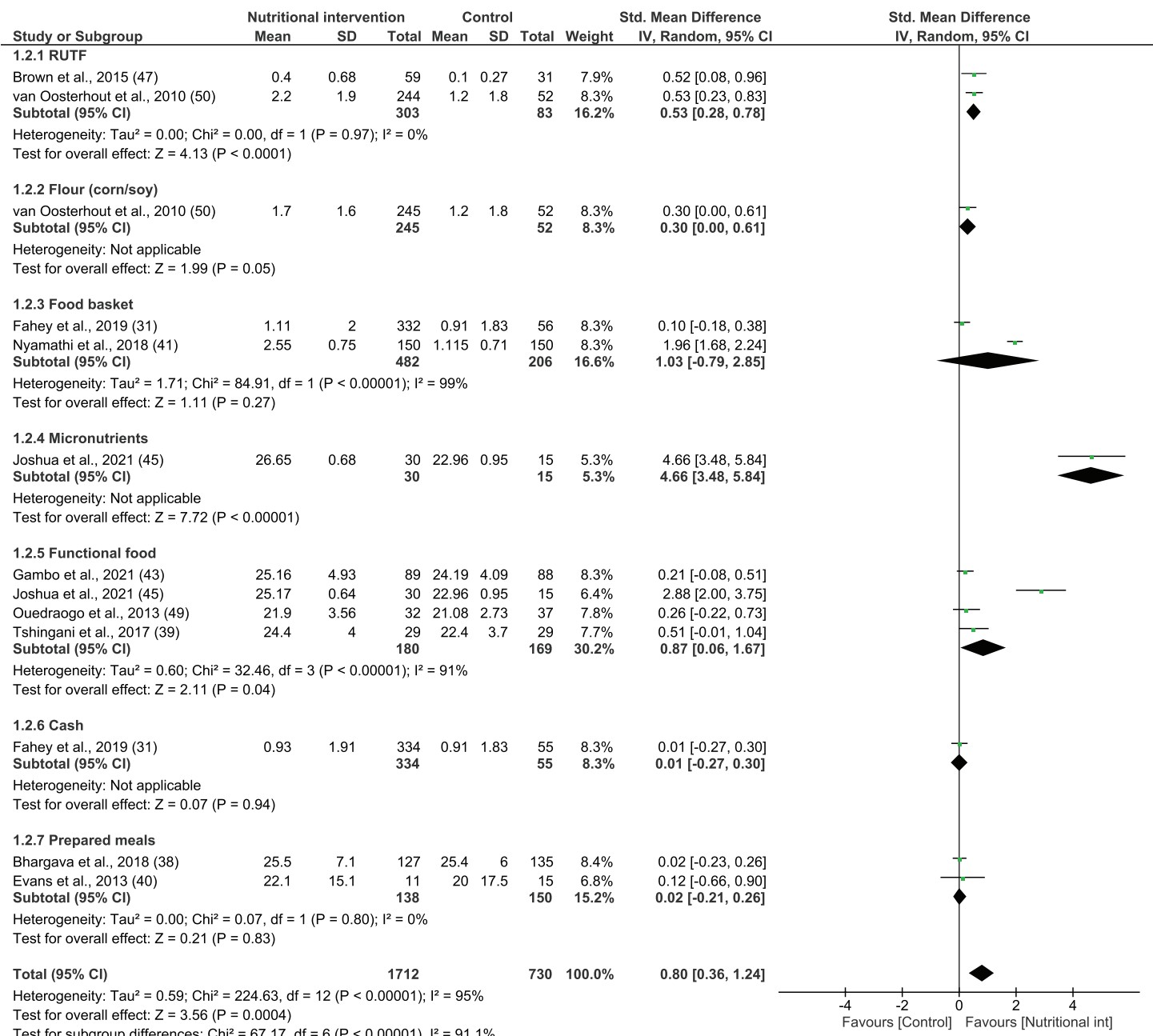

**Fig 3. Forest plot of SMD analysis results of the effect of nutritional interventions on body mass index among people living with HIV/AIDS.**

CI: 0.22, 0.59; P < 0.01; n = 720; test for heterogeneity: I² = 0%, P = 0.66) and lower-middle-income countries (SMD 1.34; 95% CI: 0.53, 2.15; P < 0.01; n = 1434; test for heterogeneity: I² = 97%, P < 0.01) significantly increased BMI among PLWHA, while studies in upper-middle-income countries showed no effect on BMI (S9 Fig).

**Effects of nutritional intervention on body composition.** The effect of nutritional interventions on the body composition of the participating PLWHA was examined in four studies that reported on three specific interventions [32,35,40,42]. Significantly improved body composition was observed in the case of women who were supplied food baskets

comprising high-protein dals (lentils, black grams, and pigeon peas) over a period of 6 months [42]. The women living with HIV/AIDS who either received only the food basket or received food baskets along with nutritional education achieved a mean (±SD) fat-free mass and a mean (±SD) fat mass of 38.1 (±7.0) kg and 14.3 (±6.4) kg, respectively, compared to the 35.5 (±6.5) kg and 12.9 (±5.2) kg mass attained by participants in the control group who received only nutritional education (p < 0.05). Furthermore, the studies conducted by Olsen et al. [35] and Yilma et al. [32], both of which reported on body composition or the proportion of lean and fat mass, found no difference between the effects of early supplementation of lipid-based nutrient supplements compared to delayed supplementation, with the mean (±SD) fat mass change in the supplemented group and the delayed-supplemented group being 1.73 (±11.10) kg and 0.52 (8.54) kg, respectively [32]. In addition, the change in mean (±SD) fat-free mass was 0.91 (±4.42) kg in the supplemented group compared to 0.06 (±3.19) kg in the nonsupplemented group [35]. Similarly, Evans et al. did not find any significant effect of FutureLife porridge® nutritional supplement (388 kcal/day) on the body composition of PLWHA [40].

**Effects of nutritional intervention on body composition: Meta-analysis.** The meta-analysis of the effect of nutritional interventions on body fat mass was performed considering three studies [32,40,42]. The combined effect of nutritional interventions on the fat mass of PLWHA compared to those who did not receive such interventions was found to be significant (3 interventions; SMD 0.21; 95% CI: 0.07, 0.34; P = 0.002; n = 908; test for heterogeneity: $I^2$ = 0%, P = 0.60) (Fig 4). In particular, this effect on fat mass was observed largely based on the findings of Carpenter et al.—a study that examined the impact of providing protein-rich food baskets to PLWHA. Notably, this study was ranked as having a low RoB, and it contributed to 66% of the sample size [42]. However, subgroup analysis by type of nutritional supplementation could not be conducted due to the presence of only one study for each type of nutritional intervention. In addition, all studies included in the meta-analysis were ranked as having a low RoB. Furthermore, two of the interventions lasted 4 to 6 months [40,42], and one lasted less than 4 months [32]. Among these, the interventions that lasted 4 to 6 months were

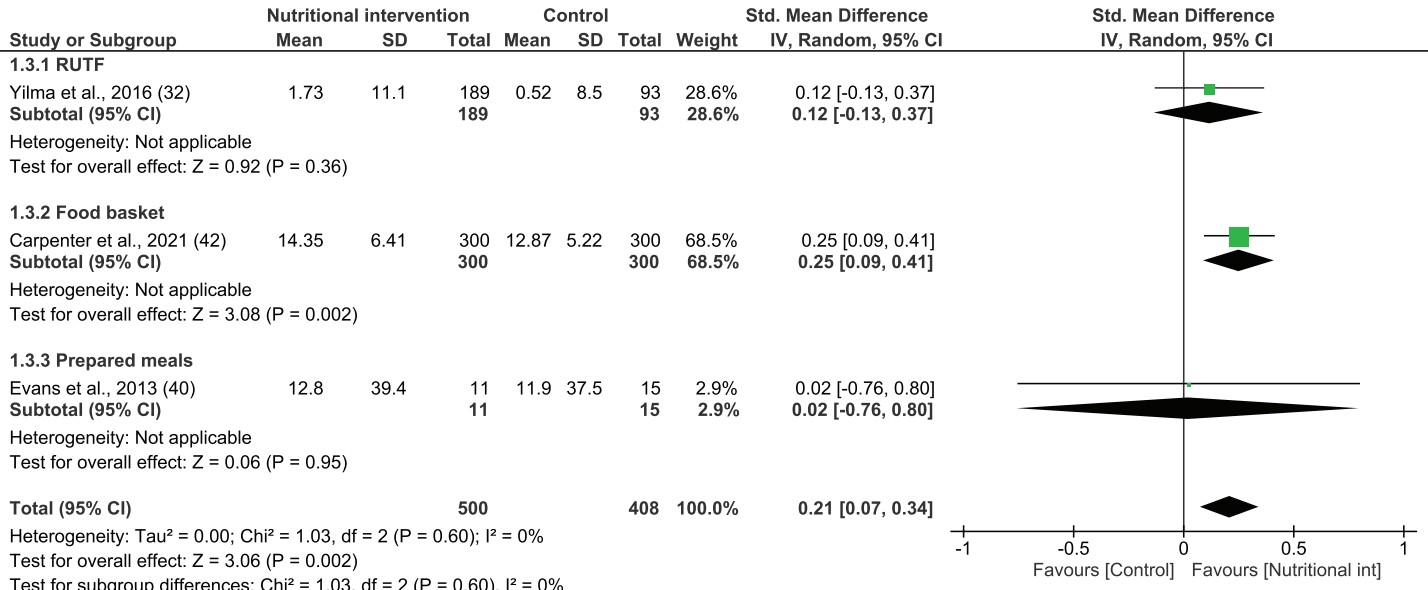

**Fig 4. Forest plot of SMD analysis results of the effect of nutritional interventions on body fat mass among people living with HIV/AIDS.**

found to have a significant effect on the fat mass of PLWHA (SMD 0.24; 95% CI: 0.09, 0.40; P = 0.002; n = 626; test for heterogeneity: $I^2$ = 0%, P = 0.57) (see S10 Fig).

Similarly, the meta-analysis pertaining to lean body mass was performed on three studies [35,40,42] to find that nutritional supplementation had a statistically significant combined effect on fat-free mass (3 interventions; SMD 0.33; 95% CI: 0.19, 0.46; P < 0.00001; n = 908; test for heterogeneity: $I^2$ = 0%, P = 0.37) (Fig 5). Furthermore, since heterogeneity was found to be null, it was not possible to carry out subgroup analysis based on the type of nutritional supplementation. All three studies included in this meta-analysis were ranked as having low RoB. Notably, the interventions that lasted 4 to 6 months had a significant effect on the fat-free mass of the PLWHA (SMD 0.37; 95% CI: 0.21, 0.53; P < 0.01; n = 626; test for heterogeneity: $I^2$ = 0%, P = 0.43) (see S11 Fig) [40,42].

**Effects of nutritional intervention on mid-upper arm circumference.** The three studies that investigated the effects of nutritional interventions on MUAC were included in this systematic review [33,47,49]. In a study conducted by Brown et al., PLWHA who received ready-to-use therapeutic food exhibited a 0.2 (±1.9) cm decline in MUAC compared to the baseline, in contrast to the slight increase in MUAC among PLWHA who did not receive any such supplements (0.04 ± 0.3 cm). However, this difference was not significant (p = 0.39) [47]. As for the other two studies, one provided optimized meals (354.92 kcal/d; soya bean, millet, moringa, and carrot) to PLWHA for 6 months [33] while the other supplied spirulina (10 g daily) for 9 months [49]. Notably, these two studies reported the final MUAC attained by the participants, while Brown et al. focused only on the change between the baseline and the endline [47]. The former studies reported a positive and statistically significant effect of nutritional interventions on MUAC. PLWHA who received optimized meals had a MUAC of 31.0 (±0.3) cm at baseline, which increased to 31.8 (±0.7) cm after 6 months. In contrast, those in the control group had a mean MUAC of 29.7 (±0.3) cm at baseline and 29.0 (±0.7) cm at the endline [33]. Furthermore, in the case of the PLWHA who received spirulina, the mean MUAC increased from 24.8 (±2.7) cm at baseline to 27.4 (±2.2) cm at the 9-month follow-up, while it increased from 24.5 (±2.4) cm to only 25.9 (±2.4) cm in the control group [49].

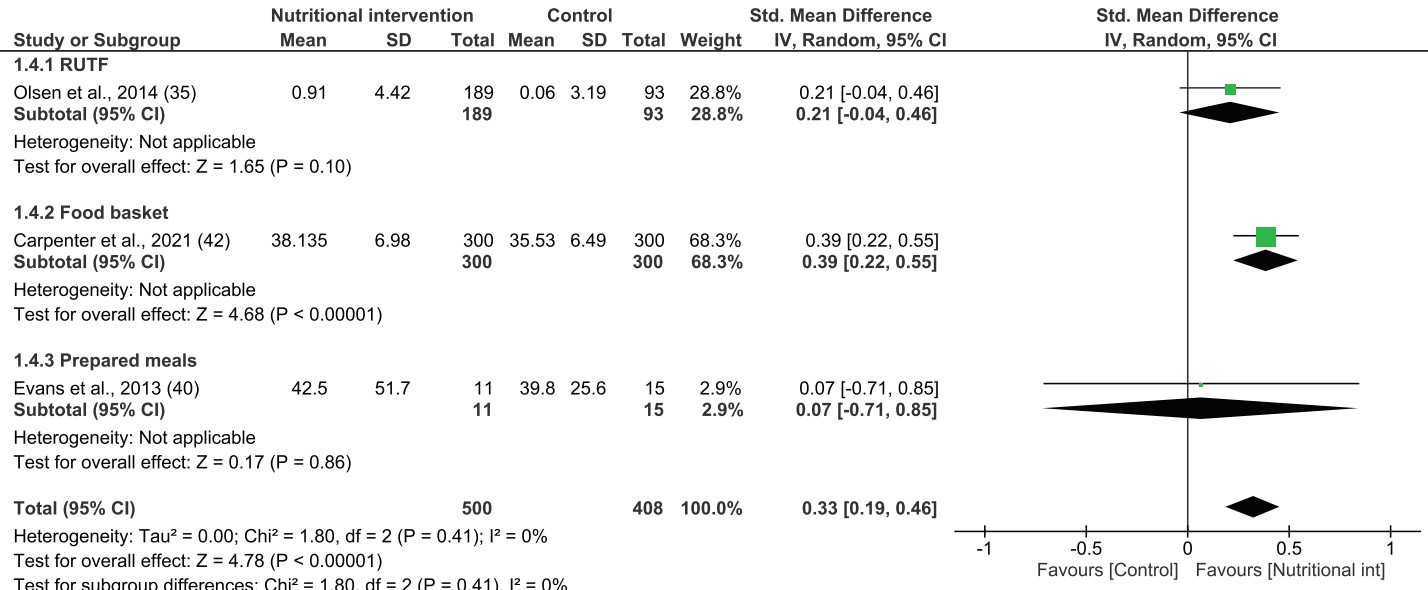

**Fig 5. Forest plot of SMD analysis results of the effect of nutritional interventions on body lean mass among people living with HIV/AIDS.**

**Effects of nutritional intervention on MUAC: Meta-analysis.** The meta-analysis of the effect of nutritional interventions for PLWHA on MUAC included three studies [33,47,49]. The combined effect of ready-to-use supplementary foods [47], optimized meals [33], and spirulina [49] on MUAC was not significantly different from that observed for the control group that received no supplementation (3 interventions; SMD 1.42; 95% CI: -0.35, 3.19; P = 0.12; n = 202; test for heterogeneity: $I^2$ = 96%, P < 0.00001) (Fig 6). Notably, the two studies that identified a significant effect of the nutritional interventions on MUAC achieved a significant increase in MUAC (SMD, 95% CI) by 3.92 (95% CI: 2.95, 4.90) cm [33] and 0.67 (95% CI: 0.18, 1.15) cm [49].

Because of the limited number of studies involved, subgroup analysis by type of nutritional supplementation could not be conducted for this factor. Furthermore, all three studies secured a high RoB. Subgroup analysis by country income status showed no significant effect among participants from lower-middle-income countries, while there was insufficient evidence for low-income and upper-middle-income countries (see S12 Fig).

**Effect of nutritional intervention on hemoglobin concentrations.** The effect of nutritional intervention on hemoglobin concentrations was reported in only four of the selected studies [36,40,44,51]. In this context, PLWHA who received multivitamin supplementation, including vitamins B complex, C, and E, for 18 months exhibited an increase in their hemoglobin concentrations to 1.0 (±1.5) g/dL compared to 0.9 (±1.7) g/dL in the placebo group (P = 0.977). PLWHA undergoing highly active antiretroviral therapy (HAART), who received 200 mg of moringa supplements for 3 months, also displayed improved hemoglobin concentrations compared to the participants in the control group (13.15 (±1.33) g/dL and 11.84 (±2.69) g/dL, respectively, P < 0.05). Furthermore, women living with HIV/AIDS who received food baskets of high-protein dals had higher hemoglobin concentrations after 6 months of intervention compared to those who received standard or nutritional education [51]. The consumption of nutritional porridge supplements for 6 months significantly improved hemoglobin concentrations in PLWHA compared to the control group, with the reported change (%) in concentration between the baseline and at

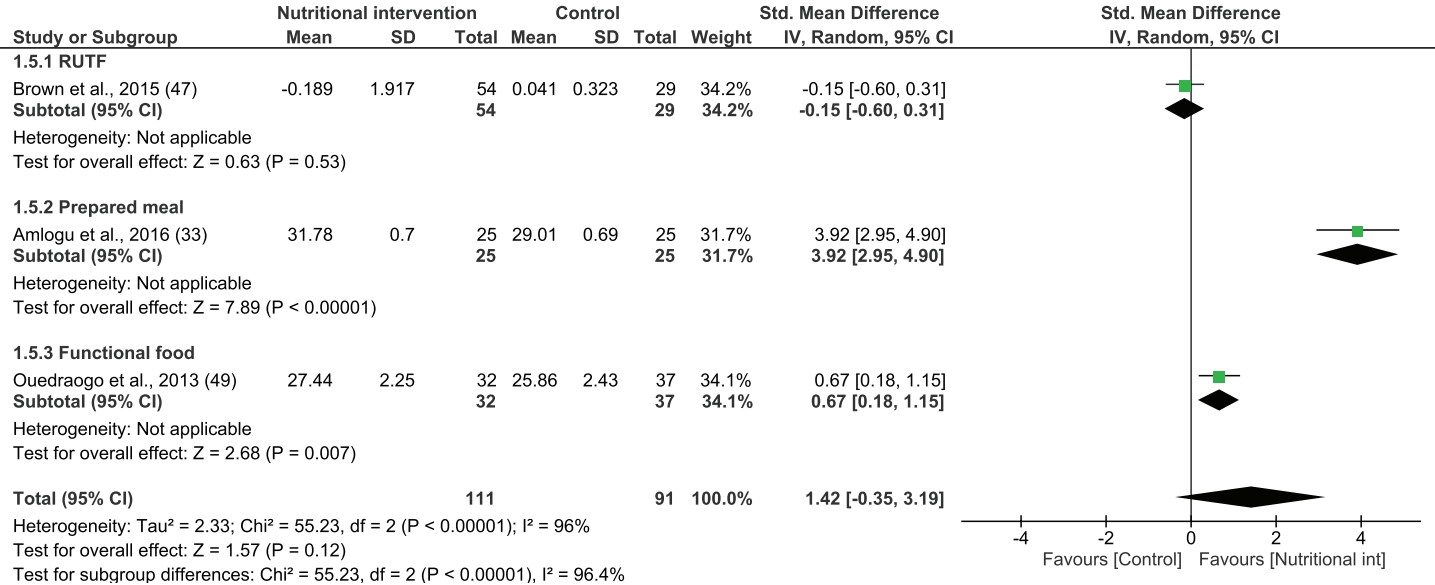

**Fig 6. Forest plot of SMD analysis results of the effect of nutritional interventions on mid-upper arm circumference among people living with HIV/AIDS.**

6-month follow-up being 9.5% (IQR 1.1–40.4) as opposed to 1.0% (IQR -1.5–7.6) for the control group (P = 0.03) [40].

**Effect of nutritional intervention on hemoglobin concentrations: Meta-analysis.** Overall, the meta-analysis performed on the four studies [36,40,44,51] showed no significant effect of nutritional interventions on the hemoglobin concentrations of PLWHA compared to the control group (4 interventions; SMD 0.46; 95% CI: -0.22, 1.13; P = 0.19; n = 1097; test for heterogeneity: I² = 96%, P < 0.00001) (Fig 7). As indicated above, the PLWHA who received nutritional porridge for 6 months exhibited a higher percentage change in hemoglobin concentration compared to the control group [40]. Since changes in the mean and SD were not reported in these studies, we used the endline results for the meta-analysis. Baseline hemoglobin concentrations between the intervention and control groups were significantly different, while study heterogeneity was not statistically significant. All included studies were ranked as having a low RoB. Furthermore, one of the interventions lasted less than 4 months [44], two lasted between 4 and 6 months [40,51], and one lasted more than 6 months [36]. Among these, the interventions administered for a duration of 4 to 6 months had no effect on the hemoglobin concentrations of PLWHA (SMD 0.56; 95% CI: -0.70, 1.82; P = 0.38; n = 626; test for heterogeneity: I² = 90%, P < 0.01) (see S13 Fig). Nutritional interventions for PLWHA in lower-middle-income countries significantly increased hemoglobin concentration (SMD 0.91; 95% CI: 0.39, 1.43; P < 0.01; n = 704; heterogeneity: I² = 83%, P = 0.02). However, no significant effects were observed in low-income and upper-middle-income countries (see S14 Fig).

**Effect of nutritional intervention on micronutrient concentrations.** Three studies examined the effects of nutritional interventions on micronutrient levels. Opara et al. reported

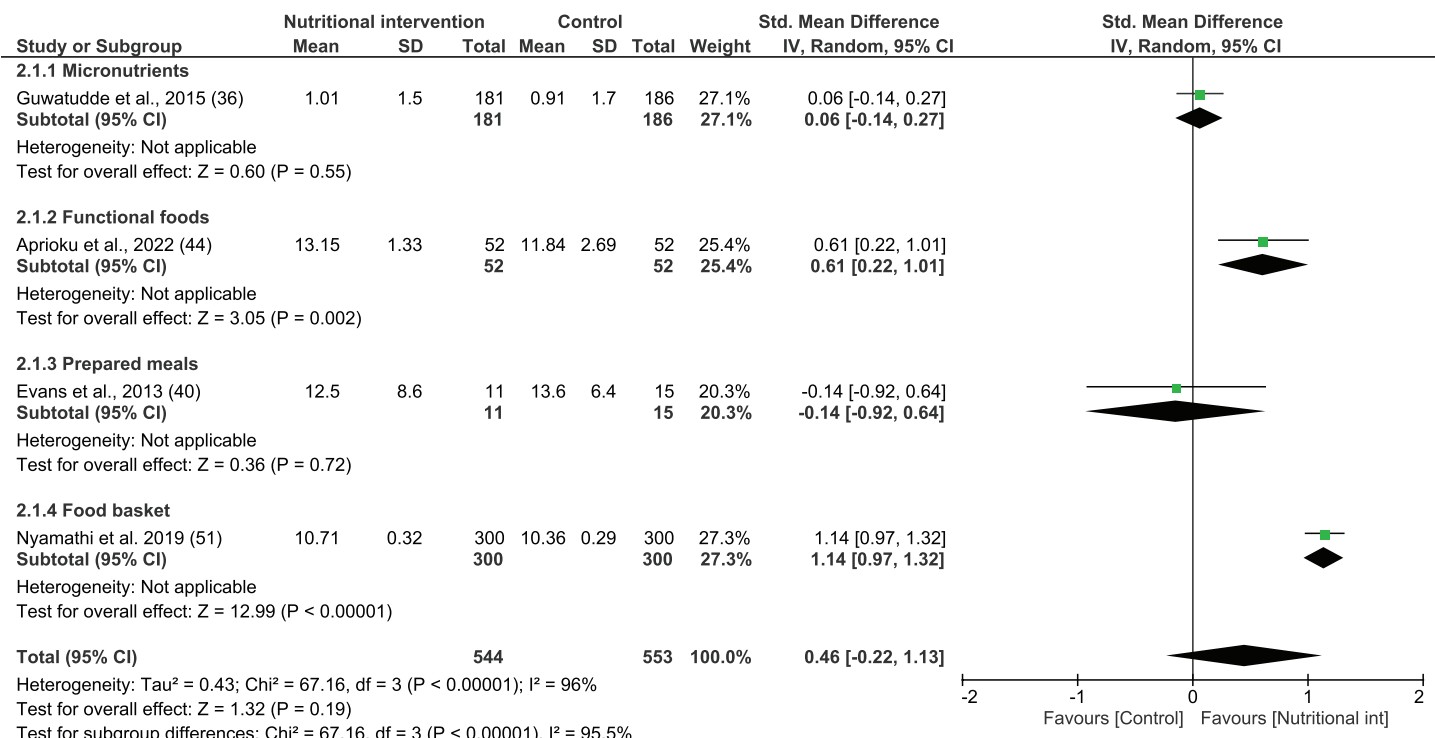

**Fig 7. Forest plot of SMD analysis results of the effect of nutritional interventions on hemoglobin concentrations among people living with HIV/AIDS.**

significant increases in packed cell volume, serum retinol, and serum vitamin C levels among participants receiving nutritional counseling and micronutrient supplements compared to controls [37]. Similarly, patients receiving a carrot-ginger blend or the commercial SelACE® supplement showed significant improvements in vitamin A (as β-carotene) and zinc levels compared to those receiving only ART without additional supplementation [45]. However, Evans et al. found no significant effect of nutritional supplementation on iron concentration [40].

**Effect of nutritional intervention on immunological status.**  This study also assessed the effects of nutritional interventions on CD4 and viral load.

**Effect of nutritional intervention on CD4.**  The 11 studies that reported the effect of nutritional interventions on CD4 were included in this systematic review [33,35,36,38–41,43–45,48]. Notably, the CD4 results of these studies were inconsistent. In the case of two interventions that provided food baskets [41,48], two studies that involved *Moringa oleifera* leaf powder supplementation [43,44], and one study that provided prepared meals [33], a significant improvement in CD4 was observed in the intervention groups compared to the controls. In particular, the effects reported by Joshua et al., who provided study participants with micronutrients rich in antioxidants (vitamins A, C, and E) and natural extracts of carrot–ginger [45], were greater than those observed in the other studies. Moreover, the effects of lipid-based nutrient supplements on immune recovery, including CD4, were observed only in the group that received whey-based supplements (compared to delayed supplementation) [35]. Notably, this systematic review also examined the combined groups that were provided nutritional supplements (whey- and soy-based protein).

Furthermore, the provision of prepared meals was found to significantly improve CD4 levels at 6 months compared to the control group [33]. However, in this case, the baseline levels of CD4 in the intervention and control groups were different—418.5 (±68.7) cells/microL and 351.8 (±62.5) cells/microL, respectively—but no statistical analysis was conducted to verify whether this difference was significant. Nonetheless, the study reported a 12.1% increase in the mean CD4 count in the intervention group at 6 months, at the same time as this value decreased by 12% in the control group. Notably, the heterogeneity of the studies included in this meta-analysis was high, with one of the studies that identified a significant effect of the interventions on CD4 featuring high RoB [48] and another presenting RoB concerns [33].

**Effect of nutritional intervention on CD4: Meta-analysis.**  Similar to the systematic review, 11 studies were included in the meta-analysis [33,35,36,38–41,43–45,48]. Overall, the nutritional interventions significantly improved CD4 levels compared to the controls (11 interventions; SMD 0.99; 95% CI: 0.46, 1.51; P = 0.0002; n = 2133; test for heterogeneity: $I^2$ = 96%, P < 0.00001) (Fig 8). Among these studies, Gambo et al. calculated the monthly change in CD4 between the intervention and placebo groups. Furthermore, supplementation with moringa was found to increase CD4 levels compared to the placebo group at the end of the 6-month period of supplementation [43]. In addition, Aprioku et al. estimated an improved CD4 count in PLWHA undergoing HAART after 3-month supplementation with moringa compared to the group that received no such supplementation [44].

Food baskets resulted in a significant improvement in CD4 compared to the controls (2 interventions; SMD 1.42; 95% CI: 0.84, 2.00; P < 0.0001; n = 750; test for heterogeneity: $I^2$ = 92%, P = 0.0005). In contrast, micronutrient supplementation (SMD 5.04; 95% CI: -5.05, 15.13; P = 0.33; n = 412; test for heterogeneity: $I^2$ = 99%, P < 0.01), functional food (SMD 0.59; 95% CI: -0.09, 1.26; P = 0.09; n = 384; test for heterogeneity: $I^2$ = 89%, P = 0.01), and prepared meals (SMD 0.61; 95% CI: -0.78, 2.00; P = 0.39; n = 338; test for heterogeneity: $I^2$ = 95%, P < 0.01) showed no significant effect on the CD4 count. In addition, no evidence was found regarding the effect of ready-to-use therapeutic food on the CD4 count of PLWHA.

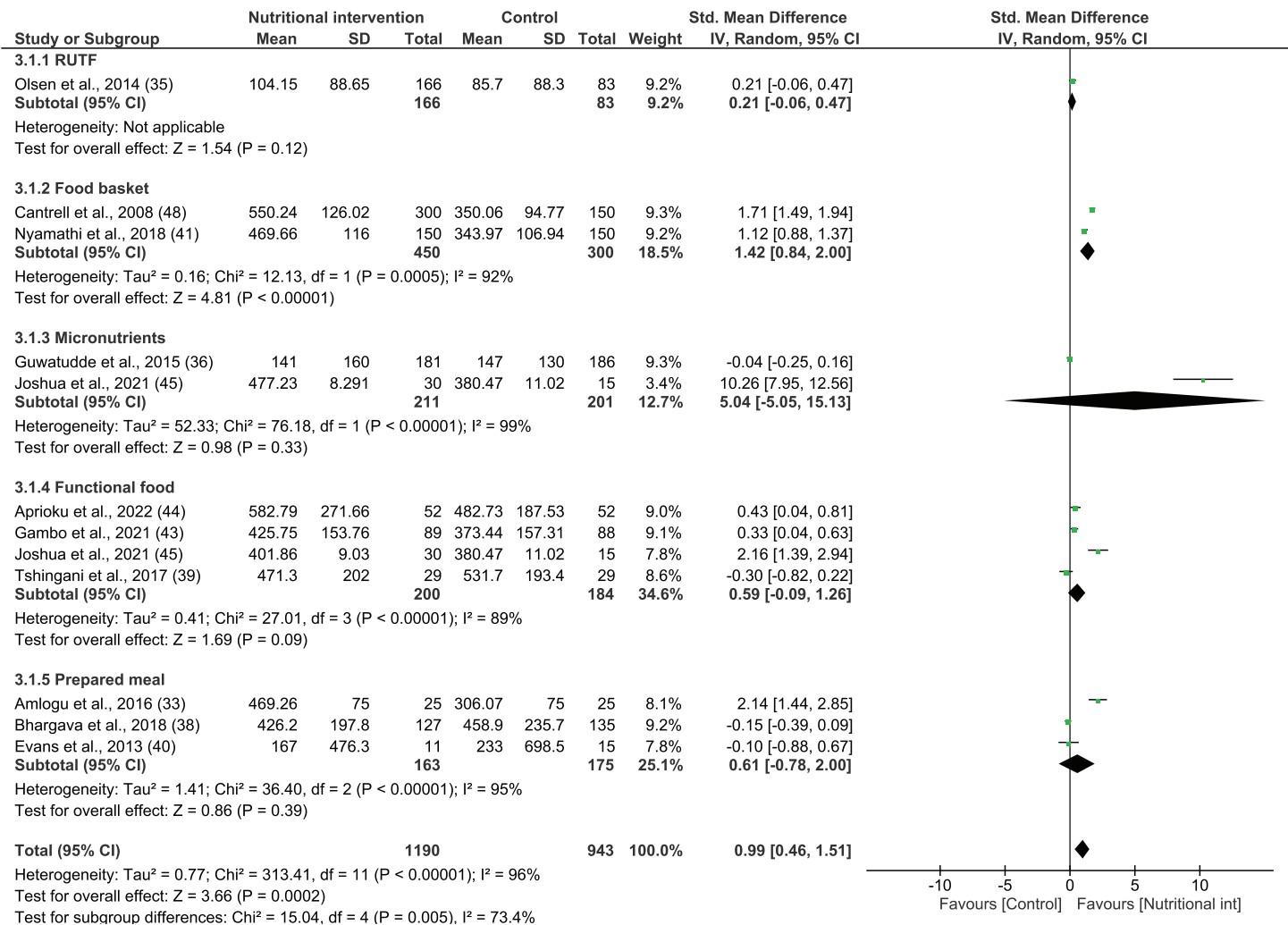

**Fig 8. Forest plot of SMD analysis results of the effect of nutritional interventions on CD4 among people living with HIV/AIDS.**

Three studies that comprised participants with a low baseline CD4 did not find any significant effect of the nutritional interventions on CD4 compared to the control groups [35,36,40]. To examine this phenomenon further, we conducted a sensitivity analysis by excluding studies with a low baseline CD4. The results displayed similar findings for both the overall nutritional intervention and for prepared meals in the sensitivity analysis compared to the full sample analysis. When considered together, the nutritional interventions significantly improved CD4 levels compared to that in the control group (SMD 1.39; 95% CI: 0.71, 2.08; P = 0.0001; n = 1,491; test for heterogeneity: $I^2$ = 97%, P < 0.00001). In contrast, prepared meals failed to significantly improve CD4 levels compared to the control group (SMD 0.97; 95% CI: -1.27, 3.22; P = 0.40; n = 312; test for heterogeneity: $I^2$ = 97%, P < 0.01).

Among the 11 studies examined, seven were found to have low RoB, three had high RoB, and one was characterized by moderate RoB. The interventions demonstrated a significant effect on CD4 levels of PLWHA in the studies with a low RoB (SMD 1.00; 95% CI: 0.41, 1.58; P < 0.01; n = 1313; test for heterogeneity: $I^2$ = 95%. P < 0.01). However, this effect was not observed in studies with a high RoB (SMD 0.52; 95% CI: -0.66, 1.70; P = 0.38; n = 370; test for heterogeneity: $I^2$ = 95%, P < 0.01) (see S15 Fig). With regard to the duration of intervention,

studies spanning less than 4 months had a significant effect on the CD4 of PLWHA (SMD 2.48; 95% CI: 1.01, 3.95; p < 0.01, n = 443; test for heterogeneity: $I^2$ = 97%, P < 0.01). In contrast, interventions lasting between 4–6 months (SMD 0.64; 95% CI: -0.05, 1.32; P = 0.07; n = 611; test for heterogeneity: $I^2$ = 92%, P < 0.01) and more than 6 months (SMD 0.51; 95% CI: -0.67, 1.68; P = 0.40; n = 1,079; test for heterogeneity: $I^2$ = 99%, P < 0.01) had no such effect (see S16 Fig). Based on the income status of the countries, studies in lower-middle-income countries showed a significant effect on the CD4 count of PLWHA (SMD: 1.84; 95% CI: 1.12, 2.55; P < 0.01; n = 1171; heterogeneity: $I^2$ = 96%, P < 0.01). However, interventions in low-income (SMD: 0.01; 95% CI: -0.22, 0.25; P = 0.93; n = 674; heterogeneity: $I^2$ = 47%, P = 0.15) and upper-middle-income countries (SMD: -0.15; 95% CI: -0.38, 0.09; P = 0.22; n = 288; heterogeneity: $I^2$ = 0%, P = 0.91) showed no significant effect on CD4 counts (see figure in S17 Fig).

**Effect of nutritional intervention on viral load.** The two studies that examined the effect of nutritional interventions on the viral load of PLWHA were included in this systematic review [35,38], along with one study that estimated the fixed effects using linear mixed models [43]. Olsen et al. found no significant difference in the suppression of viral load between PLWHA who received lipid-based nutrient supplements and those who did not [35]. Meanwhile, Bhargava et al. reported a significant decrease in mean plasma HIV RNA levels in all participants—those receiving adherence support and those receiving adherence support as well as food baskets—considering the difference between the baseline (Survey 1) and endline (Survey 3) [38]. However, the researchers could not compare the outcomes of these two groups, since the control group in this study was the one that did not receive any adherence support. Furthermore, Gambo et al. did not find any significant effect of 6-month supplementation with *Moringa oleifera* leaf powder on viral load (β = -0.005, p = 0.956) [43].

**Effect of nutritional intervention on viral load: Meta-analysis.** The viral load meta-analysis was performed on both studies examined in the systematic review above [35,38]. The combined effect of the nutritional interventions was not significant (2 interventions; SMD 3.32; 95% CI: -3.20, 9.83; P = 0.32; n = 511; test for heterogeneity: $I^2$ = 100%, P < 0.00001) (Fig 9). Notably, this analysis allowed for a comparison between the control group receiving adherence support and the intervention group, which received additional meat-based prepared meals [38]. The findings showed that plasma HIV RNA levels were higher in the intervention group (124 ± 8.1 copies/mL vs. 73 ± 7.2 copies/mL in the control group, p < 0.0001). Notably, this

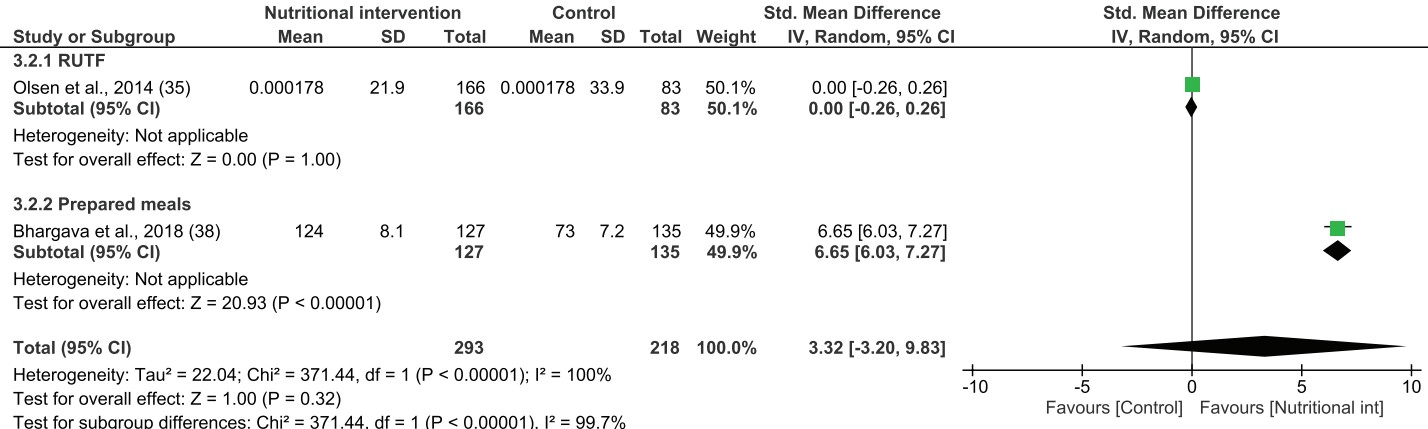

**Fig 9. Forest plot of SMD analysis results of the effect of nutritional interventions on viral load among people living with HIV/AIDS.**

analysis included the endline data reported in Survey 3. However, the baseline viral load in the intervention group was higher than in the control group (10301 ± 46 in the intervention group compared to 6836 ± 43 in the control group) [38]. This is consistent with the lower number of patients with undetectable plasma HIV RNA levels (≤25 copies/mL) in the intervention group (28.4 ± 45.2%) compared to the control group (33.3 ± 47.3%). Notably, the heterogeneity of the studies was high, while the study by Bhargava et al. was found to have a high RoB [38].

**Effect of nutritional intervention on adherence to ART among PLWHA.** Adherence to ART was measured in terms of the medication possession ratio (MPR) or stopping ART by investigating four studies that reported on five different interventions. The nutritional interventions provided in these studies were lipid-based nutrient supplements [50], CSB [50], food baskets [30,48], conditional cash [30], and prepared meals [40]. Among these studies, only Cantrell et al. noted a significant increase in ART adherence among food-insecure PLWHA who received monthly rations of micronutrient-fortified CSB and vegetable oil compared to the control group [48]. Two-thirds (70%) of the PLWHA in the food group achieved an MPR of at least 95% compared to 48% attained by the control group. Furthermore, only one participant who received lipid-based nutrient supplements for 14 weeks reported stopping ART (0.4%), while three participants who received CSB (1%) reported stopping ART, compared to 9% of the participants in the control group [50]. Meanwhile, Martinez et al. used three measures of adherence—missed clinic appointments, delayed prescription refills, and self-reported missed ART doses—to conduct a comparison between PLWHA who received nutritional education (control) and those who also received food baskets along with nutritional education [46]. Adherence in both groups improved over the course of 12 months (p < 0.01), especially within the first 6 months. At 6 months, the intervention group saw a 19.6% improvement in on-time refills compared to the control group (p < 0.01), with no significant differences observed in the other measures.

**Effect of nutritional intervention on adherence to ART among PLWHA: Meta-analysis.** The effect of nutritional interventions on adherence to ART was examined based on three studies that reported on four interventions. The overall effect was not significant, achieving an estimated risk ratio of 1.17 (95% CI: 0.99, 1.38, P = 0.06, n = 1358; test for heterogeneity: I² = 82%, P = 0.001) (Fig 10). Furthermore, the studies were statistically heterogeneous, with two of the three studies identified as presenting some RoB concerns [30,48]. Nutritional interventions for PLWHA in low-income countries showed no effect on adherence to ART (see figure in S18 Fig).

**Effect of nutritional interventions on food insecurity – Systematic review and meta-analysis.** Although food insecurity was assessed by the majority of the studies, the effect of nutritional interventions on food insecurity was reported in only three studies. Tesfaye and collaborators reported a 14.3% reduction in severe food insecurity among PLWHA receiving lipid-based nutritional supplements or those in the control group. Notably, the proportion of individuals without food insecurity increased by 15% in the intervention group but remained unchanged in the control group [34]. Fahey et al. found no significant change in the household food insecurity access scores of PLWHA who received cash, food baskets, or standard care [31]. However, severe food insecurity decreased within the groups that received cash (from 41.3% at baseline to 11.5% at 6 months of intervention) and food baskets (from 43.4% to 10.4%), in stark contrast to the plight of the control group (baseline severe food insecurity was 36.9% compared to ~29.8% at endline). Furthermore, Bhargava et al. did not find any difference in household food insecurity access scores at the baseline and endline for either the intervention group or control group [38]. Nonetheless, a significant reduction in food insecurity was observed among PLWHA with BMI < 25 kg/m² who had received nutritional supplements.

Overall, this meta-analysis emphasizes that the effect of nutritional intervention on food insecurity among PLWHA was not significant (SMD -0.06: 95% CI: -0.21, 0.10: P = 0.48: n = 1039; test for heterogeneity: I² = 0%: P = 0.72) (Fig 11).

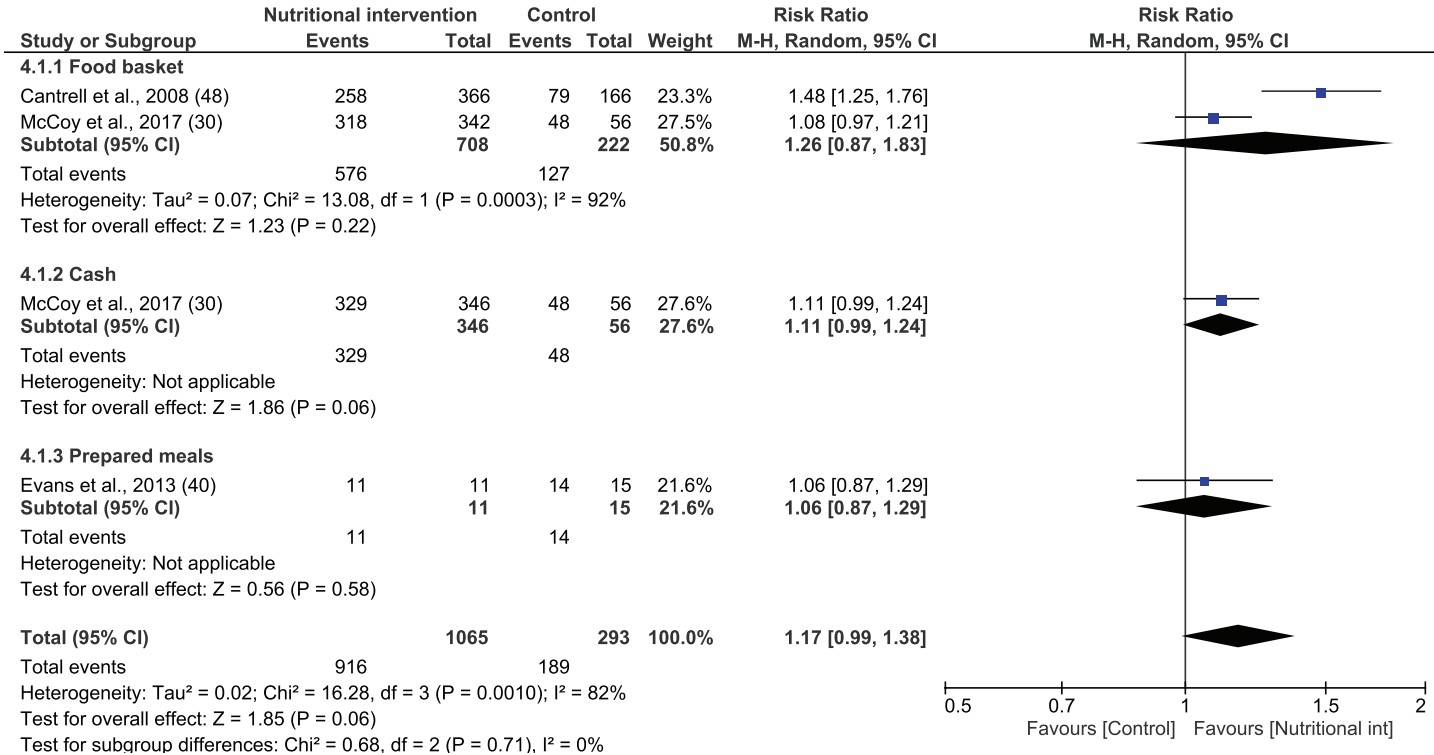

**Fig 10. Forest plot of risk ratio analysis results of the effect of nutritional interventions on adherence to ART among people living with HIV/AIDS.**

**Fig 11. Forest plot of SMD analysis results of the effect of nutritional interventions on food security among people living with HIV/AIDS.**

## Summary of the evidence

Table 4 summarizes the effects of nutritional interventions on different nutritional, immunological, and adherence outcomes.

## Discussion

The systematic reviews and meta-analyses conducted in this research highlight that nutritional interventions for PLWHA had no effect on improving weight, MUAC, hemoglobin concentration, viral load, adherence to ART, or food insecurity compared to the controls. However, they significantly improved the BMI, fat mass, fat-free mass, and CD4 of PLWHA compared to the controls. Notably, although most studies were ranked as having a low RoB, their heterogeneity resulted in achieving a moderate level of reliable evidence. Systematic reviews and meta-analyses were conducted on 22 articles that involved the provision of ready-to-use therapeutic foods, forms of lipid-based nutrient supplements, corn-soy blend, food baskets, prepared meals, micronutrients, and functional foods, including spirulina and moringa, to patients.

Furthermore, subgroup analyses were carried out when possible. Notably, differences in the study populations at baseline (e.g., their nutritional status—malnourished vs. well-nourished, biochemical parameters, HIV stage, CD4, viral load, HIV treatment, and duration of ART at the beginning of an intervention), as well as in supplement use, energy density, nutrient quality, and duration of intervention, contributed to modifying the responses to nutritional interventions.

**Table 4. Summary of the effects of nutritional interventions on nutritional, immunological, adherence, and food insecurity outcomes.**

| Outcome | Type of nutritional intervention | | | | | | | Overall effect |
|---|---|---|---|---|---|---|---|---|
| | Lipid-based nutrient supplements | Corn–soy blend | Food basket | Micro-nutrients | Functional foods | Cash | Prepared meals | |
| **Nutritional status** | | | | | | | | |
| Weight gain | ☑ | ☒ | ☐ | ☒ | ☒ | ☒ | ☐ | ☐ |
| Body mass index | ☑ | ☒ | ☐ | ☒ | ☑ | ☒ | ☐ | ☑ |
| Fat mass | ☒ | ☒ | ☒ | ☒ | ☒ | ☒ | ☒ | ☑ |
| Fat-free mass | ☒ | ☒ | ☒ | ☒ | ☒ | ☒ | ☒ | ☑ |
| Mid-upper arm circumference | ☒ | ☒ | ☒ | ☒ | ☒ | ☒ | ☒ | ☐ |
| **Micronutrient status** | | | | | | | | |
| Hemoglobin concentration | ☒ | ☒ | ☒ | ☒ | ☒ | ☒ | ☒ | ☐ |
| **Immunological status** | | | | | | | | |
| CD4 | ☒ | ☒ | ☑ | ☐ | ☐ | ☒ | ☐ | ☑ |
| Viral load | ☒ | ☒ | ☒ | ☒ | ☒ | ☒ | ☒ | ☐ |
| **Adherence** | | | | | | | | |
| Adherence to ART | ☐ | ☒ | ☒ | ☒ | ☒ | ☒ | ☒ | ☐ |
| **Secondary outcomes** | | | | | | | | |
| Food insecurity | ☒ | ☒ | ☒ | ☒ | ☒ | ☒ | ☒ | ☐ |

☐ No significant effect; ☑ Significant effect; ☒ Insufficient studies or no studies

## Lipid-based nutrient supplements led to weight gain and improved the BMI of PLWHA

The prognosis for PLWHA naïve to ARV treatment usually shows improvements for those who experience weight gain during the initial stages of ARV [35,53–55], while even 5% weight loss in 6 months markedly increases the risk of death [56]. Notably, all studies included in this systematic review and meta-analysis reported that PLWHA receiving nutritional interventions gained weight. In malnourished PLWHA, mortality varied between 5% and 8% [50] on the initiation of ART, while a low BMI increased their risk of death from two to three-fold [57,58].

Subgroup analysis based on the type of nutritional intervention was conducted for weight gain and change in BMI. Compared to corn–soy blend [50], lipid-based nutrient supplements resulted in weight gain among PLWHA to a greater extent [35,50]. Notably, lipid-based nutrient supplements are an energy dense (~1100 kcal) micronutrient paste made using peanuts, sugar, milk powder (whey or soya were also used in some formulations), oil, vitamins, and minerals, provided as single-dose sachets that can be consumed without prior preparation. This supplement has been widely used for the community treatment of severe acute malnutrition in children. The other nutritional interventions examined in this research had no significant effect on weight gain, except for the sizable effect observed in women living with HIV/AIDS who received protein-rich food baskets [41]. However, this finding should be interpreted with caution. Notably, van Oosterhout et al. included wasted participants with a mean BMI of 16.5 (±1.5) kg/m$^2$ [50] in their study, while Olsen et al. included participants with a mean BMI of 19.9 (±2.3) kg/m$^2$ [35]. A comparable pattern was identified in these studies when analyzing the effect of nutritional interventions on BMI with regard to the type of nutritional intervention, with a significantly higher effect observed for lipid-based nutrient supplements. However, both studies were conducted on malnourished PLWHA [47,50].

## Some interventions that provided food baskets improved the weight and BMI of PLWHA

The World Food Program by the United Nations sets up food basket programs in regions facing emergencies and civil or food insecurity to prevent or treat acute malnutrition, especially among children. Notably, a food basket is a virtual container that commonly consists of cereals, legumes, and oil (and sometimes sugar or micronutrients). Overall, the studies reviewed in this research highlight that compared to the control groups, food baskets had no significant effect on weight gain and BMI among PLWHA. However, in Nyamathi et al., where food baskets were provided to women with normal average BMI (20.1 ± 4.2 kg/m2) and CD4 (447.4 ± 273.6 cells/μL) [41] over a period of 6 months, an improvement in BMI was observed (SMD 1.96; 95% CI: 1.68, 2.24). In contrast, in another trial that included both men and women with normal BMI and low CD4, providing food baskets did not result in a significant improvement in BMI (SMD 0.10; 95% CI: -0.18, 0.38) [31].

These findings imply that other factors may have contributed to this disparity in the effects of similar interventions, such as the level of food insecurity and food sharing. Every participant who was offered food baskets through the interventions experienced food insecurity. According to Fahey et al., 41.6% of the PLWHA who participated in their study faced severe food insecurity, while Martinez et al. documented a higher estimation of 65%, indicating severe food insecurity among PLWHA [31,46]. Meanwhile, Cantrell et al. did not specify the percentage of severe food insecurity observed in their study [48]. Food baskets usually include balanced or protein-rich ingredients and, in most cases, are provided by accounting for household members. Despite this additional consideration, it must be noted that sharing is an important cultural and vital coping strategy for the poor, since it ensures reciprocity when

in need. For instance, during an anticipated nutritional emergency, ready-to-use supplementary foods were distributed to children younger than 5 years in Niger [59], while protection rations, including cereals, pulses, and oil, were distributed in the control areas. It was found that the child was the only one to consume supplements in the intervention areas in only 18% of the cases. This number declined to only 0.4% of the participants in the control areas. This implies the prevalence of substantial sharing within the family, along with < 15% sharing occurring outside the household, with neighbors who were not included in the distribution. In another randomized trial that provided small quantities of lipid-based nutrient supplements to 9–18-month-old children in Burkina Faso, sharing was observed for one-third of the children and in 50% of the cases [60].

## Change in weight was expressed as increase in fat mass and fat-free mass

Lean mass is an independent predictor of disease progression and mortality in PLWHA undergoing ARV treatment [35,61–63]. Weight gain during infections can be expressed in terms of fat mass, which may sustain the status of inflammation, thus highlighting the importance of determining the effects of nutritional interventions on the body composition of PLWHA. In this research, it was noted that nutritional interventions increased both fat mass and fat-free mass, although no significant difference between the two effects could be observed. Notably, subgroup analysis could not be performed for these factors due to the small number of studies. Moreover, this positive effect was observed mainly due to the findings of the study conducted by Carpenter et al., in which protein-rich food baskets were provided to women [41,42]. In addition, although the heterogeneity of these studies was null, the number of studies was too limited to draw reliable conclusions.

## Nutritional interventions had no effect on adherence to ART

PLWHA in LMICs are at a higher risk of poverty, food insecurity, and losing their source of income. These factors constitute additional risks that compel them to skip routine visits. Moreover, food insecurity may aggravate the side effects of the disease, consequently having a negative impact on ART adherence [31]. In this regard, it has been hypothesized that improving food security through cash transfers or food baskets may improve routine care attendance and adherence to ART [31].

According to the findings of this research, nutritional interventions had no effect on adherence to ART, but they successfully reduced the risk of loss to follow-up among PLWHA. Furthermore, subgroup analysis was possible only in the case of adherence to ART. While Cantrell et al. observed a significant increase in adherence to ART [48], McCoy et al. detected a significant decline in the risk of loss to follow-up, but observed no effect on adherence to ART [30]. Considering the same trial, Fahey et al. reported that the improvement in attending medical visits was not mediated by food security [31]. Furthermore, the temporal changes observed in the food insecurity scores were inconsistent. This difference in food insecurity was not significant for PLWHA who received nutritional interventions compared to those in the control group.

## Insufficient evidence of the effects of nutritional interventions on immunological markers of PLWHA

Viral load and CD4 are important biomarkers of the progress of HIV. In this research, it was observed that nutritional interventions improved CD4 levels in PLWHA but had no effect on viral load. In this context, functional foods, such as spirulina and moringa, have been widely promoted in the treatment of severe acute malnutrition in children and in the care of PLWHA. However, the number of studies conducted on this factor was too limited to provide

strong evidence. For instance, while PLWHA who received moringa exhibited improved CD4 levels in the study conducted by Gambo et al. [43], this effect was not significant in the findings of Tshingani et al. [39]. Moreover, neither study assessed the viral load.

## Strengths and limitations

The systematic review and meta-analyses investigated evidence from more than 20 studies that reported on a variety of nutritional interventions for PLWHA in LMICs. Among these, 11 were found to have a low RoB, and the majority assessed most of the outcomes considered in this research. Nonetheless, these studies are limited by their high heterogeneity and the variety of supplements, doses, duration of intervention, time since ART initiation, nutritional and immunological status of PLWHA participants, and the income status of the countries. Furthermore, the nutritional interventions administered in these studies were provided for a relatively short period, and the duration might need to be prolonged to either trigger or potentially sustain an effect in the long term. Additionally, the heterogeneity of interventions, settings, and population characteristics restricted the feasibility of conducting robust subgroup analyses, making it challenging to clearly define target groups for nutritional interventions. Furthermore, the lack of separately reported outcomes in most studies hindered gender-based analyses, limiting our understanding of the differential impacts of interventions across genders. However, the sub-group analysis conducted based on the income status in the study country showed varied results depending on the outcome.

Overall, the main analysis showed that no specific nutritional intervention was significantly superior to the controls in the context of weight gain, which is considered the main outcome of nutritional care for PLWHA. It is noteworthy that when analyzing the effects of nutritional interventions based on the income status of the country, PLWHA residing in low-income countries experienced significantly greater weight improvement compared to controls. This finding suggests that alternative indictors, such as BMI, body composition, and CD4 count, may be more appropriate for evaluating the effectiveness of nutritional interventions in this context. Therefore, further analyses and studies should be conducted to identify the optimal timing of nutritional interventions and the groups at risk that should be targeted for effective care.

## Conclusion

Although nutritional interventions, mainly lipid-based nutrient supplements and food baskets, were effective in improving some of the nutritional and immunological indicators of PLWHA, they did not have a significant impact on weight gain or adherence to ART. Moreover, the type and duration of the nutritional interventions, and the income status of the country modified the extent of the impact of the nutritional interventions. These factors as well as the immunological, nutritional, and food security status of the patients, should be accounted for with regard to the specific contexts of PLWHA before scaling up nutritional interventions. More research employing stronger and more harmonized designs should be conducted to find evidence and offer necessary guidance for establishing context-specific robust nutritional care policies for PLWHA.

## Supporting information

**S1 Checklist. PRISMA 2020 checklist.**
(DOCX)

**S1 Table. MEDLINE search strategy for the effects of nutritional interventions on the nutritional status and health of people living with HIV/AIDS.**
(DOCX)

**S2 Table. Embase search strategy for the effects of nutritional interventions on the nutritional status and health of people living with HIV/AIDS.**
(DOCX)

**S3 Table. Cochrane search strategy for the effects of nutritional interventions on the nutritional status and health of people living with HIV/AIDS.**
(DOCX)

**S4 Table. Scopus search strategy for the effects of nutritional interventions on the nutritional status and health of people living with HIV/AIDS.**
(DOCX)

**S5 Table. Web of Science search strategy for the effects of nutritional interventions on the nutritional status and health of people living with HIV/AIDS.**
(DOCX)

**S6 Table. Characteristics of the studies excluded from the systematic review and meta-analyses.**
(DOCX)

**S1 Fig. Risk of bias of the included studies, calculated using the RoB 2 tool.**
(TIF)

**S2 Fig. Forest plot of SMD analysis results of the effect of nutritional interventions on weight gain among people living with HIV/AIDS (subgroup analysis by type of nutritional intervention).**
(TIF)

**S3 Fig. Forest plot of SMD analysis results of the effect of nutritional interventions on weight gain among people living with HIV/AIDS (subgroup analysis by risk of bias).**
(TIF)

**S4 Fig. Forest plot of SMD analysis results of the effect of nutritional interventions on weight gain among people living with HIV/AIDS (subgroup analysis by duration of nutritional intervention).**
(TIF)

**S5 Fig. Forest plot of SMD analysis results of the effect of nutritional interventions on weight gain among people living with HIV/AIDS (subgroup analysis by the income status of the country).**
(TIF)

**S6 Fig. Forest plot of SMD analysis results of the effect of nutritional interventions on the body mass index of people living with HIV/AIDS (subgroup analysis by type of nutritional intervention).**
(TIF)

**S7 Fig. Forest plot of SMD analysis results of the effect of nutritional interventions on body mass index among people living with HIV/AIDS (subgroup analysis by risk of bias).**
(TIF)

**S8 Fig. Forest plot of SMD analysis results of the effect of nutritional interventions on the body mass index of people living with HIV/AIDS (subgroup analysis by duration of nutritional intervention).**
(TIF)

**S9 Fig. Forest plot of SMD analysis results of the effect of nutritional interventions on the body mass index of people living with HIV/AIDS (subgroup analysis by the income status of the country).**
(TIF)

**S10 Fig. Forest plot of SMD analysis results of the effect of nutritional interventions on body fat mass of people living with HIV/AIDS (subgroup analysis by duration of nutritional intervention).**
(TIF)

**S11 Fig. Forest plot of SMD analysis results of the effect of nutritional interventions on body fat-free mass of people living with HIV/AIDS (subgroup analysis by duration of nutritional intervention).**
(TIF)

**S12 Fig. Forest plot of SMD analysis results of the effect of nutritional interventions on MUAC of people living with HIV/AIDS (subgroup analysis by the income status of the country).**
(TIF)

**S13 Fig. Forest plot of SMD analysis results of the effect of nutritional interventions on hemoglobin concentrations among people living with HIV/AIDS (subgroup analysis by duration of nutritional intervention).**
(TIF)

**S14 Fig. Forest plot of SMD analysis results of the effect of nutritional interventions on hemoglobin concentrations among people living with HIV/AIDS (subgroup analysis by the income status of the country).**
(TIF)

**S15 Fig. Forest plot of SMD analysis results of the effect of nutritional interventions on CD4 among people living with HIV/AIDS (subgroup analysis by risk of bias).**
(TIF)

**S16 Fig. Forest plot of SMD analysis results of the effect of nutritional interventions on CD4 among people living with HIV/AIDS (subgroup analysis by duration of nutritional intervention).**
(TIF)

**S17 Fig. Forest plot of SMD analysis results of the effect of nutritional interventions on CD4 among people living with HIV/AIDS (subgroup analysis by the income status of the country).**
(TIF)

**S18 Fig. Forest plot of SMD analysis results of the effect of nutritional interventions on adherence to ART among people living with HIV/AIDS (subgroup analysis by the income status of the country).**
(TIF)

## Author contributions

**Conceptualization:** Alain Nahaskida, Stefaan De Henauw, Souheila Abbeddou.

**Data curation:** Alain Nahaskida, Jérome W. Somé, Befikadu Tariku Gutema, Nele S. Pauwels, Souheila Abbeddou.

**Formal analysis:** Alain Nahaskida, Befikadu Tariku Gutema, Souheila Abbeddou.

**Funding acquisition:** Alain Nahaskida.

**Investigation:** Alain Nahaskida, Souheila Abbeddou.

**Methodology:** Alain Nahaskida, Befikadu Tariku Gutema, Souheila Abbeddou.

**Resources:** Stefaan De Henauw, Souheila Abbeddou.

**Software:** Alain Nahaskida, Befikadu Tariku Gutema, Nele S. Pauwels, Souheila Abbeddou.

**Supervision:** Nele S. Pauwels, Souheila Abbeddou.

**Validation:** Alain Nahaskida, Jérome W. Somé, Nele S. Pauwels, Souheila Abbeddou.

**Visualization:** Alain Nahaskida, Jérome W. Somé, Befikadu Tariku Gutema, Souheila Abbeddou.

**Writing – original draft:** Alain Nahaskida, Souheila Abbeddou.

**Writing – review & editing:** Alain Nahaskida, Jérome W. Somé, Befikadu Tariku Gutema, Nele S. Pauwels, Stefaan De Henauw, Souheila Abbeddou.

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
