## [Decision Letter · Decision Letter 0]

4 Nov 2024

PONE-D-24-28337Effects of nutritional interventions on nutritional and immunological status and adherence to antiretroviral treatment among adults living with HIV in low- and middle-income countries: Systematic Review and Meta-AnalysisPLOS ONE

Dear Dr. Nahaskida,

Thank you for submitting your manuscript to PLOS ONE. After careful consideration, we feel that it has merit but does not fully meet PLOS ONE’s publication criteria as it currently stands. Therefore, we invite you to submit a revised version of the manuscript that addresses the points raised during the review process.

Please make sure that the all points raised by the reviewers are addresses in your discussion.

We look forward to receiving your revised manuscript.

Kind regards,

Anete Trajman

Academic Editor

PLOS ONE

Journal requirements: When submitting your revision, we need you to address these additional requirements. 1. Please ensure that your manuscript meets PLOS ONE's style requirements, including those for file naming. The PLOS ONE style templates can be found at https://journals.plos.org/plosone/s/file?id=wjVg/PLOSOne_formatting_sample_main_body.pdf and https://journals.plos.org/plosone/s/file?id=ba62/PLOSOne_formatting_sample_title_authors_affiliations.pdf 2. We suggest you thoroughly copyedit your manuscript for language usage, spelling, and grammar. If you do not know anyone who can help you do this, you may wish to consider employing a professional scientific editing service.  The American Journal Experts (AJE) (https://www.aje.com/) is one such service that has extensive experience helping authors meet PLOS guidelines and can provide language editing, translation, manuscript formatting, and figure formatting to ensure your manuscript meets our submission guidelines. Please note that having the manuscript copyedited by AJE or any other editing services does not guarantee selection for peer review or acceptance for publication.  Upon resubmission, please provide the following: The name of the colleague or the details of the professional service that edited your manuscript A copy of your manuscript showing your changes by either highlighting them or using track changes (uploaded as a *supporting information* file) A clean copy of the edited manuscript (uploaded as the new *manuscript* file)”.  3. Thank you for stating the following financial disclosure:  [This work was conducted under the PhD studies of AN, whose scholarship was funded by the International Atomic Energy Agency (IAEA, Vienna)].  Please state what role the funders took in the study.  If the funders had no role, please state: ""The funders had no role in study design, data collection and analysis, decision to publish, or preparation of the manuscript."" If this statement is not correct you must amend it as needed. Please include this amended Role of Funder statement in your cover letter; we will change the online submission form on your behalf. 4. As required by our policy on Data Availability, please ensure your manuscript or supplementary information includes the following:  A numbered table of all studies identified in the literature search, including those that were excluded from the analyses.   For every excluded study, the table should list the reason(s) for exclusion.   If any of the included studies are unpublished, include a link (URL) to the primary source or detailed information about how the content can be accessed.  A table of all data extracted from the primary research sources for the systematic review and/or meta-analysis. The table must include the following information for each study:  Name of data extractors and date of data extraction  Confirmation that the study was eligible to be included in the review.   All data extracted from each study for the reported systematic review and/or meta-analysis that would be needed to replicate your analyses.  If data or supporting information were obtained from another source (e.g. correspondence with the author of the original research article), please provide the source of data and dates on which the data/information were obtained by your research group.  If applicable for your analysis, a table showing the completed risk of bias and quality/certainty assessments for each study or outcome.  Please ensure this is provided for each domain or parameter assessed. For example, if you used the Cochrane risk-of-bias tool for randomized trials, provide answers to each of the signalling questions for each study. If you used GRADE to assess certainty of evidence, provide judgements about each of the quality of evidence factor. This should be provided for each outcome.   An explanation of how missing data were handled.   This information can be included in the main text, supplementary information, or relevant data repository. Please note that providing these underlying data is a requirement for publication in this journal, and if these data are not provided your manuscript might be rejected.  

Additional Editor Comments:

Please revise based on reviewers' comments.

Reviewers' comments:

Reviewer's Responses to Questions

**Comments to the Author**

1. Is the manuscript technically sound, and do the data support the conclusions?

Reviewer #1: Yes

Reviewer #2: Yes

2. Has the statistical analysis been performed appropriately and rigorously? 

Reviewer #1: Yes

Reviewer #2: Yes

3. Have the authors made all data underlying the findings in their manuscript fully available?

Reviewer #1: Yes

Reviewer #2: Yes

4. Is the manuscript presented in an intelligible fashion and written in standard English?

Reviewer #1: Yes

Reviewer #2: Yes

5. Review Comments to the Author

Reviewer #1: Line 138: what are controlled trials? arent they already included under RCTs?

Line 139: need to mention definition/criteria of LMIC used.

Line 143: What about studies which included both children and adults and also reported subgroup data for adults?

For PRISMA, the 2020 PRISMA figure for databases+other sources should be used.

Reviewer #2: Dear Editor,

I appreciate the opportunity to review the manuscript, "Effects of nutritional interventions on nutritional and immunological status and adherence to antiretroviral treatment among adults living with HIV in low- and middle-income countries: systematic review and meta-analysis." In addition to contributing to the worldwide literature and raising intriguing ideas, the article aids in the achievement of Sustainable Development Goal (SDG) goal 3.3, which calls for the eradication of the HIV epidemic by 2030.Before endorsing for publication, I have included some of my thoughts and recommendations that may strengthen the article.

Abstract

- Although the abstracts were written well, the authors should have included a few of the meta-analysis models and techniques they employed for evaluating the outcome variable.

- The study's focus is on low- and middle-income countries, but it's also important to see how many of these countries are included in the abstract (or results) section.

Introduction

-The introduction was well-written and presented; however, some of the lengthy statements needed to be supported by several citations (for e.g., lines 70-77).

Methods and Materials

- Why were pregnant and breastfeeding mothers or comorbidities excluded from the research population? Why were the authors unwilling to do sub-group analysis to determine the influence on policy? Were there any differences in the exclusion of those groups? The authors should have provided an explanation for their exclusion in the method section of their manuscript.

Result, discussion, and conclusion

- The results were well-written and narrated, and they discussed on the effect of nutritional interventions on PLWHA adherence to ART, food security, weight, BMI, body composition (including fat and fat-free mass), and MUAC. What about the findings that were made about particular countries? analysis of subgroups by gender or by country? Which countries or gender require more focus? What about gender-specific or country-specific policy implications? Are low-middle-income countries affected similarly by the findings? No environmental or geographic impact? or differences within health institutions? how the authors came up with those differences as their subject matter pertaining to LMICs.

- The recommendation is not feasible; who can benefit? low or middle-income countries?

Thanks

6. PLOS authors have the option to publish the peer review history of their article (what does this mean? ). If published, this will include your full peer review and any attached files.

**Do you want your identity to be public for this peer review?** For information about this choice, including consent withdrawal, please see our Privacy Policy .

Reviewer #1: No

Reviewer #2: No

---

## [Author Response · Author response to Decision Letter 0]

17 Jan 2025

Dear Editor

We thank the reviewers for their constructive comments, which helped us to improve our paper. Our responses to the reviewers are provided 'Response to Reviewers' as a word document.

Sincerely,

Alain Nahaskida

---

## [Decision Letter · Decision Letter 1]

10 Feb 2025

Effects of nutritional interventions on nutritional and immunological status and adherence to antiretroviral treatment among adults living with HIV in low- and middle-income countries: Systematic review and meta-analysis

PONE-D-24-28337R1

Dear Dr. Nahaskida,

We’re pleased to inform you that your manuscript has been judged scientifically suitable for publication and will be formally accepted for publication once it meets all outstanding technical requirements.

Kind regards,

Anete Trajman

Academic Editor

PLOS ONE

Additional Editor Comments (optional):

We thank the authors for revising the manuscript based on the reviewers' comments.

Reviewers' comments:

Reviewer's Responses to Questions

**Comments to the Author**

1. If the authors have adequately addressed your comments raised in a previous round of review and you feel that this manuscript is now acceptable for publication, you may indicate that here to bypass the “Comments to the Author” section, enter your conflict of interest statement in the “Confidential to Editor” section, and submit your "Accept" recommendation.

Reviewer #2: All comments have been addressed

2. Is the manuscript technically sound, and do the data support the conclusions?

Reviewer #2: Yes

3. Has the statistical analysis been performed appropriately and rigorously? 

Reviewer #2: Yes

4. Have the authors made all data underlying the findings in their manuscript fully available?

Reviewer #2: Yes

5. Is the manuscript presented in an intelligible fashion and written in standard English?

Reviewer #2: Yes

6. Review Comments to the Author

Reviewer #2: I appreciate the editor inviting me to review the manuscript again. I was grateful to the authors for their thoughtful revision and addressed all of my comments and recommendations. Lastly, in line with Sustainable Development Goal (SDG) 3.3, which calls for the eradication of the HIV pandemic by 2030, I will suggest the authors expand their discussion and offer suggestions for policymakers and healthcare professionals.

7. PLOS authors have the option to publish the peer review history of their article (what does this mean? ). If published, this will include your full peer review and any attached files.

**Do you want your identity to be public for this peer review?** For information about this choice, including consent withdrawal, please see our Privacy Policy .

Reviewer #2: **Yes: ** Habtamu Endashaw Hareru

---

## [Editor Report · Acceptance letter]

PONE-D-24-28337R1

PLOS ONE

Dear Dr. Nahaskida,

I'm pleased to inform you that your manuscript has been deemed suitable for publication in PLOS ONE. Congratulations! Your manuscript is now being handed over to our production team.

Kind regards,

on behalf of

Professor Anete Trajman

Academic Editor

PLOS ONE